# S4M: S4 FOR MULTIVARIATE TIME SERIES FORECASTING WITH MISSING VALUES

**Jing Peng**[1]  **Meiqi Yang**[2]  **Qiong Zhang**[1]*  **Xiaoxiao Li**[3,4]

[1]Renmin University of China   [2]Princeton University
[3]The University of British Columbia   [4]Vector Institute
`{jpeng, qiong.zhang}@ruc.edu.cn,   meiqiy@princeton.edu`
`xiaoxiao.li@ece.ubc.ca`

## ABSTRACT

Multivariate time series data play a pivotal role in a wide range of real-world applications, such as finance, healthcare, and meteorology, where accurate forecasting is critical for informed decision-making and proactive interventions. However, the presence of block missing data introduces significant challenges, often compromising the performance of predictive models. Traditional two-step approaches, which first impute missing values and then perform forecasting, are prone to error accumulation, particularly in complex multivariate settings characterized by high missing ratios and intricate dependency structures. In this work, we introduce S4M, an end-to-end time series forecasting framework that seamlessly integrates missing data handling into the Structured State Space Sequence (S4) model architecture. Unlike conventional methods that treat imputation as a separate preprocessing step, S4M leverages the latent space of S4 models to directly recognize and represent missing data patterns, thereby more effectively capturing the underlying temporal and multivariate dependencies. Our framework comprises two key components: the Adaptive Temporal Prototype Mapper (ATPM) and the Missing-Aware Dual Stream S4 (MDS-S4). The ATPM employs a prototype bank to derive robust and informative representations from historical data patterns, while the MDS-S4 processes these representations alongside missingness masks as dual input streams to enable accurate forecasting. Through extensive empirical evaluations on diverse real-world datasets, we demonstrate that S4M consistently achieves state-of-the-art performance. These results underscore the efficacy of our integrated approach in handling missing data, showcasing its robustness and superiority over traditional imputation-based methods. Our findings highlight the potential of S4M to advance reliable time series forecasting in practical applications, offering a promising direction for future research and deployment. Code is available at https://github.com/WINTERWEEL/S4M.git.

## 1 INTRODUCTION

Multivariate time series are common in real-world applications, including finance (Zhang et al., 2024), health care (Kaushik et al., 2020), and meteorology (Duchon & Hale, 2012). *Time series forecasting* (Box et al., 2015) predicts future values based on historical data. Accurate forecasting enables informed decision making and helps anticipate trends and take proactive measures, from optimizing financial investments to improving patient care and responding to environmental changes.

Time series forecasting has been a long-standing area of research, with numerous methods developed over the years. Traditional statistical methods typically build on linear assumptions and autoregressive models to capture temporal dependency, such as ARIMA (Box & Jenkins, 1968), failing to forecast well in complex multivariate time series. Recent machine learning advancements have introduced promising solutions, including RNN-based methods (Salinas et al., 2017; Rangapuram et al., 2018; Lim et al., 2020; Hewamalage et al., 2021) that capture long-term dependencies and attention-based models (Qin et al., 2017; Shih et al., 2019; Wu et al., 2021; Liu et al., 2022; Shabani

---

*Correspondence to: Qiong Zhang qiong.zhang@ruc.edu.cn.

et al., 2023; Nie et al., 2022; Liu et al., 2023; Xue et al., 2023) that leverage temporal attention mechanisms. A more recent and influential technique is the Structured State Space Sequence (S4) model (Gu et al., 2021), which combines the strengths of state-space models with modern deep learning architectures to efficiently model long sequences. This study highlights the strong suitability of S4 models for time-series forecasting, driven by their ability to address the growing demand for efficiency in large-scale applications where computational resources and scalability are paramount.

A significant challenge in time series forecasting is the effective handling of missing data, which often arises due to sensor failures, data collection issues, or external disruptions. Missing values can severely degrade model performance if not properly addressed. For instance, in healthcare, gaps in wearable device data may occur due to inconsistent usage (Darji et al., 2023); in financial transactions, data might be incomplete owing to network outages or system downtimes (Emmanuel et al., 2021); in environmental monitoring, sensor networks measuring air or water quality frequently face data loss due to device malfunctions or harsh weather scenarios (Zhang & Thorburn, 2022). In these applications, the data may exhibit block missing patterns, where missing values occur consecutively rather than randomly (see Fig. 5). These gaps not only reduce the amount of available data but can also introduce biases, leading to inaccurate forecasts.

Traditional approaches to handling missing data in time series typically involve a two-step process: imputing missing values and then performing forecasting on the imputed data (Cao et al., 2018; Cini et al., 2021; Marisca et al., 2022). However, in multivariate time series, the complexity of missing data patterns and high missing ratios make direct imputation challenging. This two-step approach often leads to error accumulation, resulting in suboptimal forecasting performance. Consequently, there is a growing need for end-to-end methods that integrate missing data handling directly into the forecasting process. Existing methods for handling missing data have notable limitations. RNN-based methods like GRU-D (Che et al., 2018) and BRITS (Cao et al., 2018) address missing data but often *require long training and perform poorly*. Graph models like BiTGraph (Chen et al., 2023) capture dependencies at high memory cost, while ODE-based methods like Neural ODE (Chen et al., 2018), GraFITi (Yalavarthi et al., 2024) and CRUs (Schirmer et al., 2022) are *computationally expensive*. Fig. 1 presents a comprehensive comparison of forecasting accuracy, training time, and memory usage across the methods. The results highlight that S4-based methods not only achieve superior performance but also offer significantly lower computational costs compared to alternative approaches. This efficiency underpins our decision to focus on S4 in this work. A more detailed review of related work is provided in Appendix A.

To address these challenges, we propose an *end-to-end* method in this work that is both computationally and memory efficient while maintaining robust forecasting performance under block missing data. We build on the S4 model due to its demonstrated success in time series forecasting (Wang et al., 2024) and its ability to handle multiple inputs concurrently. This capability allows us to address missing data while simultaneously learning the complex dependency structures inherent in the forecasting task. By integrating missing data handling directly into the S4 framework, we aim to fully leverage its strengths for multivariate time series forecasting.

Our method termed S4 with missing values (`S4M`) that explicitly considers missing values in the S4 model consists of two modules: adaptive temporal prototype mapper (`ATPM`) and missing-aware dual stream S4 (`MDS-S4`). The `ATPM` module is designed to use rich historical data patterns stored in a prototype bank to learn robust and informative representations of the time sequence. These representations, along with a mask indicating missing values, are then processed by the `MDS-S4` module, which

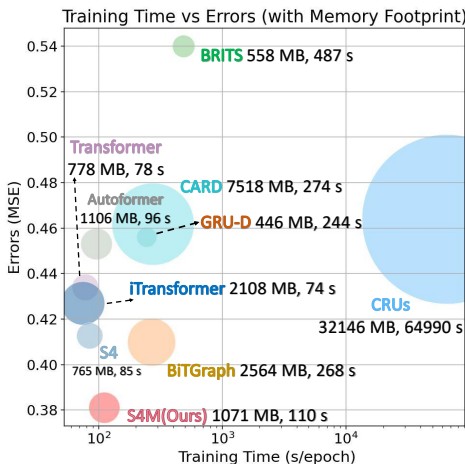

Figure 1: Comparison of prediction MSE versus training time for various methods on the Electricity dataset. Each method is represented by a dot, with size scaled according to its memory footprint. Lower values for MSE, training time, and memory indicate better performance. Our `S4M` method demonstrates superior performance across all metrics.

performs forecasting using a dual-stream S4 architecture. As shown in Fig. 1, S4M demonstrates superior efficiency in both training and inference compared to existing baselines. We conduct extensive experiments on real-world datasets, comparing our method with state-of-the-art approaches and their variants. The results demonstrate that S4M consistently achieves top-tier performance across various settings, highlighting its robustness in handling missing data. Our work addresses the critical challenge of missing data in time series forecasting, offering a scalable, efficient, and robust solution for real-world applications.

## 2 PRELIMINARY

The S4 model, introduced by Gu et al. (2021), is a pioneering sequence model designed to handle continuous-time data with *long-range dependencies*, making it highly effective for tasks like time series forecasting. For completeness, we provide a brief overview of S4.

Let $\boldsymbol{u}(t), \boldsymbol{y}(t) \in \mathbb{R}^D$ be two $D$-variate continuous signals. The continuous state space model (SSM) maps $\boldsymbol{u}(t)$ to $\boldsymbol{y}(t)$ via the following equations:

$$\frac{d}{dt}\boldsymbol{h}(t) = \boldsymbol{A}\boldsymbol{h}(t) + \boldsymbol{B}\boldsymbol{u}(t), \quad \boldsymbol{y}(t) = \boldsymbol{C}\boldsymbol{h}(t) + \boldsymbol{D}\boldsymbol{u}(t), \tag{1}$$

where $\boldsymbol{h}(t) \in \mathbb{R}^H$ is an unobserved hidden state, and the system is parameterized by matrices $\boldsymbol{A} \in \mathbb{R}^{H \times H}$, $\boldsymbol{B} \in \mathbb{R}^{H \times D}$, $\boldsymbol{C} \in \mathbb{R}^{H \times H}$, and $\boldsymbol{D} \in \mathbb{R}^{H \times D}$. Since real-world data is typically observed at discrete time points $t = 0, 1, \ldots, T$, the continuous model in equation 1 can be discretized as:

$$\boldsymbol{h}_t = \overline{\boldsymbol{A}}\boldsymbol{h}_{t-1} + \overline{\boldsymbol{B}}\boldsymbol{u}_t, \quad \boldsymbol{y}_t = \boldsymbol{C}\boldsymbol{h}_t + \boldsymbol{D}\boldsymbol{u}_t \tag{2}$$

where $\overline{\boldsymbol{A}} = (\boldsymbol{I} - \Delta\boldsymbol{A}/2)^{-1}(\boldsymbol{I} + \Delta\boldsymbol{A}/2)$ and $\overline{\boldsymbol{B}} = (\boldsymbol{I} - \Delta\boldsymbol{A}/2)^{-1}\Delta\boldsymbol{B}$ are based on bilinear transform (Gu et al., 2021) with some parameter $\Delta$. By recursively applying the recurrent representation of SSM in equation 2 model over discrete time, the output $\boldsymbol{y}_t$ at time $t$ is computed as a *convolution* of all previous inputs $u_{0:t}$:

$$\boldsymbol{y}_t = \sum_{i=0}^{t} \boldsymbol{C}\overline{\boldsymbol{A}}^{t-i}\overline{\boldsymbol{B}}\boldsymbol{u}_{t-i} + \boldsymbol{D}\boldsymbol{u}_t.$$

For an input sequence $\boldsymbol{u} = (\boldsymbol{u}_0, \boldsymbol{u}_1, \ldots, \boldsymbol{u}_T)$, one can observe that the output sequence $\boldsymbol{y} = (\boldsymbol{y}_0, \boldsymbol{y}_1, \ldots, \boldsymbol{y}_T)$ can be computed using a convolution with a skip connection $\boldsymbol{y} = \boldsymbol{C}\boldsymbol{K} * \boldsymbol{u} + \boldsymbol{D}\boldsymbol{u}$, where $*$ is the convolution operation and $\boldsymbol{K} = (\overline{\boldsymbol{B}}, \overline{\boldsymbol{A}}\overline{\boldsymbol{B}}, \ldots, \overline{\boldsymbol{A}}^{T-1}\overline{\boldsymbol{B}})$ is called the SSM kernel. One key challenge of discrete-time SSMs is that computing the output involves repeated matrix multiplications by $\overline{\boldsymbol{A}}$, which can be expensive, with a computational cost of $O(H^2T)$ when implemented naively. S4 addresses two main challenges compared to basic SSMs. First, it solves the long-range dependencies modeling challenge by employing the HiPPO matrix (Gu et al., 2020) for $\boldsymbol{A}$, enabling continuous-time memorization. Second, S4 solves the computational bottleneck by introducing a specialized representation and algorithm that significantly reduces the computational cost.

## 3 PROPOSED METHOD

### 3.1 PROBLEM FORMULATION

We denote $\boldsymbol{X}^{(L)}$ and $\boldsymbol{X}^{(H)}$ the look-back and horizon windows for the forecast, respectively, of corresponding lengths $\ell_L$ and $\ell_H$. Given a starting time $t_0$, they are denoted as $\boldsymbol{X}^{(L)} = \{\boldsymbol{x}_t \in \mathbb{R}^D : t \in t_0 : t_0 + \ell_L\}$ and $\boldsymbol{X}^{(H)} = \{\boldsymbol{x}_t : t \in t_0 + \ell_L + 1 : t_0 + \ell_L + \ell_H\}$. We consider the case where there exist missing values in the observations due to the failure of devices or some other unexpected errors. We use a mask matrix $\boldsymbol{M}^{(L)} \in \mathbb{R}^{\ell_L \times D}$ to denote whether the value is missing or not. Specifically, the $(t, d)$-th element in the mask matrix is binary and is given by

$$M_{td}^{(L)} = \begin{cases} 1, & \text{if } X_{td}^{(L)} \text{ is observed,} \\ 0, & \text{otherwise.} \end{cases}$$

The goal of forecasting is to predict the horizon window $\boldsymbol{X}^{(H)}$ given the look-back window $\boldsymbol{X}^{(L)}$. Thus, time series forecasting can be framed as learning a mapping $f$ from $\boldsymbol{X}^{(L)}$ to $\boldsymbol{X}^{(H)}$.

We design an approach to learn $f$ that is parameterized by $\boldsymbol{\theta}$ in the presence of missing data. During training, let $f(\boldsymbol{X}^{(L)}, \boldsymbol{M}^{(L)}; \boldsymbol{\theta})$ be the predicted values for the horizon window, then the parameter $\boldsymbol{\theta}$ is learned by minimizing the error between the true horizon window $\boldsymbol{X}^{(H)}$ and its predicted value. Note that the input and output of $f$ have the same length, for the foresting task where $\ell_H \leq \ell_L$, we slice the last $\ell_H$ as the predicted value.

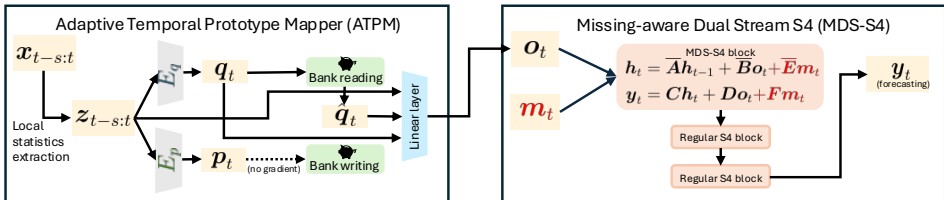

Figure 2: **Illustration of our end-to-end prediction method `S4M`**. Our method consists of two modules. The first `ATPM` module uses historical data patterns to learn robust and informative representations for the current input time sequence. Specifically, we extract the local statistics $\boldsymbol{z}_{t-s:t}$ of the time series at time point $t$ based on raw values $\boldsymbol{x}_{t-s:t}$. These statistics are then fed into the query encoder $E_q$ to obtain $\boldsymbol{q}_t$, which queries the prototype bank to retrieve the prototype $\hat{\boldsymbol{q}}_t$. Both $\boldsymbol{q}_t$ and $\hat{\boldsymbol{q}}_t$ are subsequently fed into a linear layer to produce the final representation $\boldsymbol{o}_t$. Additionally, the prototype encoder $E_p$ generates the prototype $\boldsymbol{p}_t$ for bank updating. In the second module `MDS-S4`, we model the representation $\boldsymbol{o}_t$ and the mask $\boldsymbol{m}_t$ using S4 to generate the forecast $\boldsymbol{y}_t$.

**Method Overview:** The pipeline of our proposed S4 with missing values (`S4M`) is given in Fig. 2. It consists of two modules specifically designed to deal with missing values in *an end-to-end manner*. The first `ATPM` module focuses on representation learning with missing values, it contains a *prototype bank*, which stores a rich set of representations of historical data in the time series, from which we can query the representation of missing values based on their local features. The second `MDS-S4` module directly models the missing patterns in the SSM. Our design explicitly considers missing values in the model, and the model also progressively updates the missing patterns.

### 3.2 ADAPTIVE TEMPORAL PROTOTYPE MAPPER (ATPM)

#### 3.2.1 OVERVIEW OF ATPM

To address missing values, we leverage a *prototype bank* that stores a rich set of representative patterns from time series. *The goal is to utilize historical data patterns to learn robust and informative representations for the current input time sequence.* Since the raw time series input is multivariate and can be noisy, often containing missing values, rather than querying and storing prototypes using the raw time series data, we design encoders to extract more robust latent representations, allowing us to query and store the prototypes in the representation space. As the prototypes in the bank evolve and are adaptive to the data during training, we call this module the adaptive temporal prototype mapper (ATPM).

Specifically, recall $\boldsymbol{x}_t \in \mathbb{R}^D$ is the value of the look-back window $\boldsymbol{X}^{(L)}$ at time $t$. ATPM first extracts local statistics $\boldsymbol{z}_t$ at each time point $t$ (such as its first previous non-missing value and the time difference to the first non-missing time point) based on the look-back window $\boldsymbol{X}^{(L)}$. We denote this local statistics extraction as $\boldsymbol{z}_t = f_{\text{local}}(\boldsymbol{x}_t)$, and its details are given in Appendix C.1.

At the $t$-th time point, our hypothesis is that local statistics $\boldsymbol{z}_t$ of a single time point is insufficient to infer patterns when $t$ corresponds to a missing observation. To mitigate this, we look back over a *short period* of length $s$ to assist with inference at the missing time point, constructing a matrix $\boldsymbol{z}_{t-s:t} = \{\boldsymbol{z}_l : l \in t - s : t\}$. This local statistics sequence $\boldsymbol{z}_{t-s:t}$ is then used to query and update the prototype bank in the representation space by feeding it into a query encoder $E_q$ with parameter $\boldsymbol{\theta}_q$ to obtain the query representation, which is used to query the prototype bank, and a prototype encoder $E_p$ with parameter $\boldsymbol{\theta}_p$ to obtain the prototype representation, which is used to update the prototype bank. After querying the prototype bank, we combine the retrieved prototype and other local statistics to obtain the final representation $\boldsymbol{o}_t$, which is detailed below.

### 3.2.2 DESIGN OF THE PROTOTYPE BANK

The core concept of the prototype bank is to read (query) similar representations from rich historical data stored in the bank. These representations are then used as input for the subsequent module. At the same time, the representations are also used to write (update) the bank adaptively. We describe the structure of the bank and how to read and write the bank below.

**Bank Storage.** Prototypes are organized in a two-level queue. The first level represents different clusters, with each element serving as the centroid of a cluster of prototypes. Within each cluster, the second-level queue stores the corresponding prototypes that belong to that cluster. To ensure efficient storage, inference, and stability, the first-level queue can hold a maximum of $K_1$ centroids, while each second-level queue can accommodate up to $K_2$ prototypes per cluster. The prototype bank is designed as a queue to facilitate updates following the First-In-First-Out (FIFO) principle, allowing outdated prototypes that no longer align with the updated encoder to be filtered out efficiently. The prototype bank is initialized at its first level by applying $k$-means clustering on the output of the encoder of the first batch.

**Bank Reading.** Denote $q_t = E_q(z_{t-s:t}; \theta_q)$ be the query encoder that has local temporal and spatial information. We then use $q_t$ to query the prototype bank to retrieve the most similar patterns and use their weighted average as the prototype vector at the time point $t$. In cases where $t$ is a missing value time point, the retrieved prototypes help account for the missing values. Specifically, let $\{c_1, c_2, \ldots\}$ represent the cluster centroids stored in the first-level queue, and let $q_t$ be the query feature. We compute their cosine similarity as $\rho_{tj} = q_t^\top c_j / \|q_t\| \|c_j\|$. Let $\mathbb{S}_t = \{j_1, \ldots, j_K\}$ where $\rho_{t,j_1} \geq \rho_{t,j_2} \geq \cdots \geq \rho_{t,j_K} \geq \cdots$ be the index of the top $K$ maximum similarities and normalize them as $w_{tj} = \exp(\rho_{tj}) / \sum_{j' \in S_t} \exp(\rho_{tj'})$ for $j \in \mathbb{S}_t$. These retrieved prototypes are then aggregated as: $\hat{q}_t = \sum_{j \in \mathbb{S}_t} w_{tj} c_j$. Chandar et al. (2016) observed that selecting the top $K$ similar centroids, rather than using all centroids, can improve performance. Finally, we combine $z_{t-s:t}$, $q_t$, and $\hat{q}_t$ using a dense layer to form a single representation $o_t$.

**Bank Writing.** After querying the prototype bank, we also update it using the output from $p_t = E_p(z_{t-s:t}; \theta_p)$ be the output of $E_p$. We compute the cosine similarity between this representation and the prototype centroids to assess their closeness. If the current patterns are very similar to existing prototypes, we add them to the level two queue; otherwise, we add the prototype to the level one queue as a new cluster. Specifically, let $\omega_t = \max_j p_t^\top c_j / \|p_t\| \|c_j\|$ represent the similarity value of the current representation to existing prototype centroids. If $\omega_t \geq \tau_1$ for some predefined hyper-parameter $\tau_1$, then $p_t$ is added to the queue of the cluster with which it shares the highest degree of similarity. If $\omega_t < \tau_2$ for some predefined hyper-parameter $\tau_2$, indicating insufficient similarity with any existing centroid, $p_t$ is introduced as a novel pattern to the bank and also serves as the initialization of its prototypes cluster[1]. In both cases, the centroids are updated accordingly. In the case where $\tau_1 \leq \omega_t \leq \tau_2$, the prototype is not used for updating the bank. This process ensures that the prototype bank remains dynamic and capable of capturing a diverse range of patterns.

---

**Algorithm 1** Bank Reading

**Input:** local statistics $\{z_t\}_{t=1}^{\ell_L}$, query encoder $E_q$, bank prototype centroids $\{c_1, c_2, \ldots, \}$, initial values for parameter $W$ and $d$
**Output:** target representation $O$
1: **for** $z_t$ in $Z = \{z_1, z_2, ..., z_{\ell_L}\}$ **do**
2:     **Encoding:** $q_t = E_q(z_{t-s:t})$
3:     **Similarity:** $\rho_{tj} = q_t^\top c_j / \|q_t\| \|c_j\|$
4:     **Normalization for top-K maximum values:** $w_{tj} = \exp(\rho_{tj}) / \sum_{j \in \mathbb{S}_t} \exp(\rho_{tj})$
5:     **Aggregate prototype:** $\hat{q}_t = \sum_{j \in \mathbb{S}_t} w_{tj} c_j$
6:     **Combine:** $v_t = W[z_t, q_t, \hat{q}_t] + d$
7:     **Output:** $o_t = q_t + v_t$
8: **end for**
9: **Final Output** $O = \{o_1, o_2, ..., o_{\ell_L}\}$

---

### 3.2.3 ENCODER UPDATE

Recall that the prototype $p_t = E_p(z_{t-s:t}; \theta_p)$ and the query feature $q_t = E_q(z_{t-s:t}; \theta_q)$ are the outputs of two distinct encoders, $E_q$ and $E_p$, parameterized by $\theta_p$ and $\theta_q$, respectively. The architecture of the encoders are given in Appendix C.2. Although both encoders take the same input, they serve different purposes: the prototype encoder $E_p$ is designed to store a rich set of time series

---

[1]We set $\tau_1 = 0.9$ and $\tau_2 = 0.6$ in experiment.

representations, while the query encoder $E_q$ aims to obtain a representation that diverges from the prototypes. Thus, these encoders must not be identical and should be updated differently. To ensure that the prototypes evolve more stably, we use a momentum update for the prototype encoder $E_p$, while the query encoder updates its parameters in a traditional manner. Specifically, the parameter $\theta_q$ the query encoder is updated using gradient descent based on the final loss, whereas the parameter $\theta_p$ of the prototype encoder is updated with a momentum-based approach, allowing for smoother updates as suggested by He et al. (2020). During the prototype bank writing process, the gradients of $\theta_p$ are disabled, and the parameters are updated via momentum:

$$\theta_p = \gamma\theta_p + (1 - \gamma)\theta_q \qquad (3)$$

where $\gamma \in [0, 1)$ is the momentum coefficient. The momentum update in equation 3 makes $\theta_p$ evolves more smoothly than $\theta_q$.

---

**Algorithm 2** Bank Writing

---

**Input:** local statistics $\boldsymbol{Z} = \{\boldsymbol{z}_t\}_{t=1}^{\ell_L}$, prototype encoder $E_p$, bank prototype centroids $\{\boldsymbol{c}_1, \boldsymbol{c}_2, \dots, \}$
**Output:** bank with updated prototypes
1: **Random sample** $n$ **slices** $\{\boldsymbol{z}_i\}_{i=1}^n$ from $\boldsymbol{Z}$
2: **for** $\boldsymbol{z}_i$ in $\{\boldsymbol{z}_1, \boldsymbol{z}_2, ..., \boldsymbol{z}_n\}$ **do**
3:     **Encoding:** $\boldsymbol{p}_i = E_p(\boldsymbol{z}_i)$
4:     **Similarity:** $\rho_{ij} = \boldsymbol{p}_i^\top \boldsymbol{c}_j / \|\boldsymbol{p}_i\|\|\boldsymbol{c}_j\|$
5:     **Maximum index:** $j^* = \arg\max_j \rho_{i,j}$
6:     **if** $\rho_{ij^*} \geq \tau_1$ **then**
7:         **Add** $\boldsymbol{p}_i$ to the end of the $j^*$ second-level queue
8:         **Update** $j^*$**th prototype centriod**
9:     **else if** $\rho_{ij^*} < \tau_2$ **then**
10:        **Add** $\boldsymbol{p}_i$ **to the end of the first-level queue**
11:     **else** continue
12:     **end if**
13: **end for**

---

## 3.3 MISSING-AWARE DUAL STREAM S4 (MDS-S4)

Drawing inspiration from the GRU-D model in (Che et al., 2018), we explicitly model the missing values by including the mask $\boldsymbol{M}^{(L)}$ in the SSM. Intuitively, with the presence of missing values, both the hidden state $\boldsymbol{h}_t$ and the output of S4 depend on the mask vector $\boldsymbol{m}_t$. We therefore modify the SSM so that it has two input streams: the representation and the mask. Specifically, let $\boldsymbol{o}_t$ be the output from the representation learning module, and $\boldsymbol{m}_t, \boldsymbol{y}_t$ be the $t$th row of $\boldsymbol{M}^{(L)}$ and $\boldsymbol{X}^{(H)}$. Our missing-aware dual stream SSM is:

$$\begin{aligned} \boldsymbol{h}_t &= \overline{\boldsymbol{A}}\boldsymbol{h}_{t-1} + \overline{\boldsymbol{B}}\boldsymbol{o}_t + \overline{\boldsymbol{E}}E_m(\boldsymbol{m}_t; \boldsymbol{\theta}_m) \\ \boldsymbol{y}_t &= \boldsymbol{C}\boldsymbol{h}_t + \boldsymbol{D}\boldsymbol{o}_t + \boldsymbol{F}E_m(\boldsymbol{m}_t; \boldsymbol{\theta}_m), \end{aligned} \qquad (4)$$

where $\overline{\boldsymbol{A}}$ and $\overline{\boldsymbol{B}}$ are the same as in equation 2 and $\overline{\boldsymbol{E}} = (\boldsymbol{I} - \Delta\boldsymbol{A}/2)^{-1}\Delta\boldsymbol{E}$. The encoder $E_m$ parameterized by $\boldsymbol{\theta}_m$ is used to ensure that we also use the latent representation of the mask to fully utilize its information. Denote $\boldsymbol{o} = (\boldsymbol{o}_{t_0}, \dots, \boldsymbol{o}_{t_0+\ell_L})$, $\boldsymbol{m} = (\boldsymbol{m}_{t_0}, \dots, \boldsymbol{m}_{t_0+\ell_L})$, $\boldsymbol{y} = (\boldsymbol{y}_{t_0}, \dots, \boldsymbol{y}_{t_0+\ell_L})$. Given the initial hidden state, the dual stream SSM in equation 4 can be recursively unrolled to get the following explicit convolution operation:

$$\boldsymbol{y} = \boldsymbol{C}\boldsymbol{K}_1 * \boldsymbol{o} + \boldsymbol{C}\boldsymbol{K}_2 * E_m(\boldsymbol{m}; \boldsymbol{\theta}_m) + \boldsymbol{D}\boldsymbol{o} + \boldsymbol{F}E_m(\boldsymbol{m}; \boldsymbol{\theta}_m)$$

where $\boldsymbol{K}_1 = (\overline{\boldsymbol{B}}, \overline{\boldsymbol{A}}\overline{\boldsymbol{B}}, \dots, \overline{\boldsymbol{A}}^{\ell_L-1}\overline{\boldsymbol{B}})$ and $\boldsymbol{K}_2 = (\overline{\boldsymbol{E}}, \overline{\boldsymbol{A}}\overline{\boldsymbol{E}}, \dots, \overline{\boldsymbol{A}}^{\ell_L-1}\overline{\boldsymbol{E}})$ are two SSM kernels.

Therefore, our modified SSM model for missing data has an additive structure of the SSM model in equation 2. We can use the same trick in S4 to efficiently calculate the convolution operation and end with adding two outputs from the convolution operations. The convolution operation, together with the HiPPO matrix $A$, enables S4 to effectively model long-term dependencies. Similarly, our dual-stream SSM incorporates a convolution operation and the HiPPO matrix, preserving S4's computational efficiency and capacity for modeling long-term dependencies, while simultaneously addressing missing information through distinct computational kernels. Given the output from MDS-S4, we can further feed it into either MDS-S4 or regular S4 blocks to increase

---

**Algorithm 3** Testing Pipeline

---

**Input:** Look-back window $\boldsymbol{X}^{(L)}$, learned prototype bank centroids $\mathbb{C} = \{\boldsymbol{c}_j\}$, query encoder $E_q$, learned `MDS-S4` module and local statistics extractor $f_{\text{local}}$
**Output:** Forecasted value $\hat{\boldsymbol{Y}}$
1: **Local Feature Extraction:** $\boldsymbol{Z} = f_{\text{local}}(\boldsymbol{X})$
2: **Bank Reading:** $\boldsymbol{O} = \text{Alg. 1}(\boldsymbol{Z}, \mathbb{C}, E_q)$
3: **`MDS-S4` Output:** $\hat{\boldsymbol{Y}} = \text{MDS-S4}(\boldsymbol{O})$

---

the complexity of our model. We describe the specific structure of the encoder $E_m$ and multiple S4 blocks in Appendix C.3. Our full algorithm for training and testing is, respectively, given in Alg. 4 and Alg. 3.

## 4 EXPERIMENTS

### 4.1 DATASETS AND EXPERIMENT SETUP

We select four commonly used time series datasets for forecasting: Electricity (Wu et al., 2021), ETTh1 (Zhou et al., 2021), Traffic (Wu et al., 2021), and Weather (Wu et al., 2021). Since these benchmark datasets are complete, we manually created block missing on the training and test dataset. These datasets span various domains and encompass diverse characteristics in terms of magnitude ranges, sampling frequencies, and statistical properties like seasonality. The base statistics of the data set can be found in Tab. 7. To model practical scenarios where sensors cannot record data for a period due to failure or other reasons, we design block-based missing pattern for two types of missing data scenarios: time point missing and variable missing with missing rate $r = 0.03, 0.06, 0.12, 0.14$. The details of making missing pattern can be found in Appendix D.2. After obtaining the dataset with missing values, we split it chronologically into training, validation, and test sets, with a ratio of $0.7/0.1/0.2$. The horizon window for all methods is fixed at 96, while the lookback length is varied across 96, 192, 384, and 768. In addition to these benchmark datasets, we also conduct experiments on a real-world dataset, as detailed in Appendix D.8.

---

**Algorithm 4** Training Pipeline

---

**Input:** Batches of look-back window $\{\boldsymbol{X}_i\}_{i=1}^B$ and corresponding masks $\{\boldsymbol{M}_i\}_{i=1}^B$, initial values for model parameters

**Output:** Prediction $\{\hat{\boldsymbol{Y}}_i\}_{i=2}^B$

1: **Initialize:** prototype centroids $\mathbb{C} = \{\boldsymbol{c}_1, \dots, \boldsymbol{c}_K\}$ based on $K$-means from $E_p(\boldsymbol{X}_1; \boldsymbol{\theta}_p)$
2: **for** $i = 2$ to $B$ **do**
3:     **Local Feature Extraction:** $\boldsymbol{Z}_i = f_{\text{local}}(\boldsymbol{X}_i)$
4:     **Bank Reading:** $\boldsymbol{O}_i = $ Alg. 1$(\boldsymbol{Z}_i, \mathbb{C}, E_q)$
5:     **Bank Writing (No grad):** $\mathbb{C} = $ Alg. 2$(\boldsymbol{Z}_i, \mathbb{C}, E_p)$
6:     **(No Gradient) Momentum Update:** $\boldsymbol{\theta}_p = \gamma\boldsymbol{\theta}_p + (1-\gamma)\boldsymbol{\theta}_q$
7:     **Backbone Output:** $\hat{\boldsymbol{Y}}_i = \texttt{MDS-S4}(\boldsymbol{O}_i, \boldsymbol{M}_i)$
8:     **Loss construction & backpropagation:** $\mathcal{L} = \|\hat{\boldsymbol{Y}}_i - \boldsymbol{X}_i\|_F^2$
9: **end for**

---

### 4.2 COMPETING METHODS

We compare our proposed method, S4M, with two main groups of baseline methods: S4-based baselines and other state-of-the-art and classical methods for handling missing data. The S4-based baseline group includes S4 (Mean), S4 (Ffill), S4 (Decay), and S4 (SAITS). These methods impute missing data using strategies such as global mean, last observation, a decay mechanism based on these statistics, and the superior imputation method SAITS (Du et al., 2023). The other methods include classic RNN-based methods like GRUD (Che et al., 2018), LSTM-based methods such as BRITS (Cao et al., 2018), the top-performing Transformer-based methods Transformer (Vaswani et al., 2017) and Autoformer (Wu et al., 2021), and the end-to-end method BiTGraph (Chen et al., 2023), which is specifically designed for missing data prediction.

### 4.3 COMPARISON WITH BASELINES AND S4-BASED VARIANTS ON TIME POINT MISSING

**Varying Input Length.** The results in Tab. 1 illustrate the forecasting performance of various methods under *time point missing* scenarios $r = 0.06$ across the four datasets. Our proposed S4M consistently achieves the best or second-best performance across most settings, demonstrating its robustness in handling missing data. For the Weather dataset, our method exhibits outstanding performance, achieving the best MSE in nearly all configurations, particularly at the 192-step length with 0.225, which is significantly better than the closest competitor. For the other datasets, S4M maintains strong performance, as no competing methods can consistently outperform it across various datasets and settings.

**Varying Missing Ratio.** Fig. 3 illustrates the performance of various methods under time point missing scenarios across four datasets: Electricity, ETTh1, Weather, and Traffic. The methods are evaluated using MAR as the missing ratio ($r$) increases. Across all datasets, our proposed S4M (denoted by the red line), consistently maintains lower MAE compared to other methods, particularly as the missing ratio increases. For the Electricity and Weather datasets, S4M outperforms competing methods at all missing ratios, showing a clear advantage in handling missing data. In the ETTh1 and Traffic datasets, while some other methods like GRU-D or BRITS perform well at lower missing ratios, S4M still demonstrates robust performance, particularly as $r$ increases, showing strong resilience to higher levels of missing data.

Table 1: Comparison of forecasting performance of S4M (Ours) and baselines on four datasets with various look-back window length under time point missing scenario when missing ratio $r = 0.06$. Entries with '–' indicate the experiment can not be done due to out-of-memory issue.

| Data | $\ell_L$ | Metric ↓ | BRITS | GRU-D | Trans. | Auto. | BiTGraph | S4 (Mean) | S4 (Ffill) | S4 (Decay) | S4 (SAITS) | S4M (Ours) |
|---|---|---|---|---|---|---|---|---|---|---|---|---|
| Electricity | 96 | MAE | 0.633 | 0.431 | 0.399 | 0.375 | 0.397 | 0.408 | 0.418 | 0.402 | 0.432 | **0.372** |
| | | MSE | 0.623 | 0.363 | 0.400 | **0.272** | 0.309 | 0.337 | 0.345 | 0.323 | 0.372 | 0.287 |
| | 192 | MAE | 0.636 | 0.437 | 0.402 | 0.366 | 0.388 | 0.387 | 0.384 | 0.381 | 0.394 | **0.367** |
| | | MSE | 0.628 | 0.366 | 0.314 | **0.257** | 0.290 | 0.303 | 0.292 | 0.289 | 0.309 | 0.274 |
| | 384 | MAE | 0.653 | 0.434 | 0.419 | 0.369 | 0.384 | 0.383 | **0.367** | 0.379 | 0.394 | 0.370 |
| | | MSE | 0.659 | 0.363 | 0.339 | **0.272** | 0.295 | 0.298 | 0.272 | 0.285 | 0.307 | 0.277 |
| | 768 | MAE | 0.644 | 0.437 | 0.416 | 0.379 | 0.387 | 0.378 | 0.384 | 0.379 | 0.393 | **0.373** |
| | | MSE | 0.656 | 0.365 | 0.333 | 0.285 | 0.290 | 0.291 | 0.288 | 0.285 | 0.306 | **0.282** |
| ETTh1 | 96 | MAE | 0.705 | 0.644 | 0.905 | 0.866 | **0.571** | 0.629 | 0.625 | 0.614 | 0.851 | 0.571 |
| | | MSE | 0.937 | 0.793 | 0.942 | 0.923 | **0.613** | 0.747 | 0.759 | 0.716 | 0.914 | 0.624 |
| | 192 | MAE | 0.707 | 0.653 | 0.898 | 0.797 | 0.609 | 0.600 | 0.605 | 0.595 | 0.788 | **0.574** |
| | | MSE | 0.721 | 0.805 | 0.938 | 0.885 | 0.745 | 0.670 | 0.681 | 0.666 | 0.881 | **0.593** |
| | 384 | MAE | 0.755 | 0.649 | 0.968 | 0.791 | 0.601 | 0.595 | 0.605 | 0.605 | 0.719 | **0.571** |
| | | MSE | 1.029 | 0.798 | 0.973 | 0.882 | 0.721 | 0.662 | 0.689 | 0.683 | 0.840 | **0.624** |
| | 768 | MAE | 0.788 | 0.668 | 1.110 | 0.797 | 0.599 | 0.614 | 0.614 | 0.619 | 0.733 | **0.588** |
| | | MSE | 1.072 | 0.841 | 1.041 | 0.885 | 0.684 | 0.697 | 0.710 | 0.706 | 0.848 | **0.647** |
| Weather | 96 | MAE | 0.419 | 0.363 | 0.421 | 0.465 | 0.516 | 0.371 | 0.361 | 0.399 | 0.440 | **0.313** |
| | | MSE | 0.372 | 0.293 | 0.350 | 0.395 | 0.510 | 0.312 | 0.296 | 0.344 | 0.407 | **0.237** |
| | 192 | MAE | 0.427 | 0.346 | 0.308 | 0.471 | 0.419 | 0.332 | 0.318 | 0.347 | 0.384 | **0.305** |
| | | MSE | 0.385 | 0.268 | 0.238 | 0.408 | 0.385 | 0.255 | 0.235 | 0.274 | 0.320 | **0.225** |
| | 384 | MAE | 0.434 | 0.342 | 0.391 | 0.479 | 0.587 | 0.329 | 0.345 | 0.339 | 0.378 | **0.306** |
| | | MSE | 0.375 | 0.271 | 0.310 | 0.430 | 0.596 | 0.249 | 0.269 | 0.264 | 0.311 | **0.220** |
| | 768 | MAE | 0.489 | 0.354 | 0.374 | 0.489 | 0.467 | 0.330 | 0.349 | 0.340 | 0.368 | **0.316** |
| | | MSE | 0.445 | 0.280 | 0.297 | 0.459 | 0.445 | 0.250 | 0.272 | 0.263 | 0.287 | **0.232** |
| Traffic | 96 | MAE | 0.667 | 0.467 | **0.421** | 0.430 | 0.516 | 0.455 | 0.459 | 0.451 | 0.498 | 0.428 |
| | | MSE | 1.158 | 0.871 | **0.726** | 0.812 | 0.919 | 0.808 | 0.844 | 0.794 | 0.917 | 0.809 |
| | 192 | MAE | 0.667 | 0.473 | 0.419 | 0.410 | 0.496 | 0.401 | 0.398 | 0.386 | 0.415 | **0.385** |
| | | MSE | 1.170 | 0.893 | 0.728 | 0.721 | 0.836 | 0.709 | 0.692 | 0.711 | 0.734 | **0.687** |
| | 384 | MAE | 0.675 | 0.483 | 0.452 | 0.496 | 0.527 | 0.400 | 0.398 | **0.381** | 0.412 | 0.385 |
| | | MSE | 1.193 | 0.918 | 0.746 | 0.817 | 0.913 | 0.690 | **0.682** | 0.702 | 0.711 | 0.702 |
| | 768 | MAE | 0.697 | 0.490 | 0.410 | 0.465 | – | 0.394 | 0.392 | **0.381** | 0.407 | 0.388 |
| | | MSE | 1.236 | 0.947 | 0.706 | 0.774 | – | 0.687 | **0.678** | 0.692 | 0.716 | 0.699 |

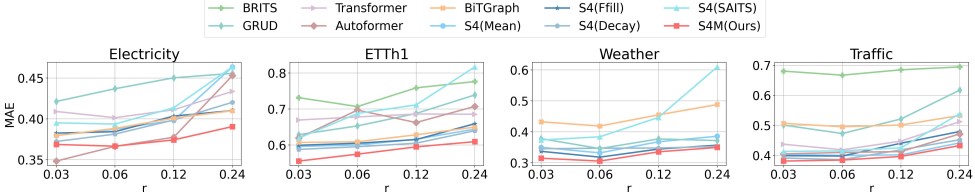

Figure 3: The performance of different methods on four datasets under time point missing scenario when the missing ratio $r$ varies from 0.03 to 0.24.

## 4.4 COMPARISON WITH BASELINES AND S4-BASED VARIANTS ON VARIABLE MISSING

**Varying Input Length.** Tab. 2 presents the forecasting performance of different methods under *variable missing* scenarios ($r = 0.06$) across four datasets. Our method, S4M, consistently achieves either the best or second-best results across the majority of configurations, demonstrating its robust-

ness in handling feature-missing data. On the ETTh1 dataset, S4M shows particularly strong results, securing the lowest MAE and MSE values in several settings. Similarly, for the Weather dataset, S4M excels, delivering the best MAE and MSE in all configurations. Across the remaining datasets, S4M continues to perform competitively, consistently matching or surpassing other methods, highlighting its general effectiveness in feature-missing scenarios.

Table 2: Comparison of forecasting performance of S4M (Ours) and baselines on four datasets with various look-back window length under variable missing scenario when missing ratio $r = 0.06$. Entries with '–' indicate the experiment can not be done due to out-of-memory issue.

| Data | $\ell_L$ | Metric↓ | BRITS | GRU-D | Trans. | Auto. | BiTGraph | S4 (Mean) | S4 (Ffill) | S4 (Decay) | S4 (SAITS) | S4M (Ours) |
|---|---|---|---|---|---|---|---|---|---|---|---|---|
| Electricity | 96 | MAE | 0.439 | 0.426 | 0.400 | 0.373 | 0.383 | 0.387 | 0.387 | 0.396 | 0.432 | **0.369** |
| | | MSE | 0.369 | 0.354 | 0.312 | 0.271 | 0.292 | 0.305 | 0.304 | 0.311 | 0.354 | 0.282 |
| | 192 | MAE | 0.457 | 0.477 | 0.400 | 0.366 | 0.376 | 0.366 | 0.365 | 0.378 | 0.405 | **0.357** |
| | | MSE | 0.390 | 0.408 | 0.308 | **0.257** | 0.277 | 0.273 | 0.272 | 0.282 | 0.310 | 0.261 |
| | 384 | MAE | 0.625 | 0.470 | 0.412 | 0.361 | 0.389 | 0.366 | 0.367 | 0.377 | 0.411 | **0.359** |
| | | MSE | 0.619 | 0.408 | 0.317 | **0.255** | 0.290 | 0.270 | 0.272 | 0.279 | 0.317 | 0.264 |
| | 768 | MAE | 0.635 | 0.487 | 0.411 | 0.363 | 0.387 | 0.367 | 0.376 | 0.374 | 0.402 | **0.362** |
| | | MSE | 0.637 | 0.434 | 0.326 | **0.261** | 0.287 | 0.272 | 0.286 | 0.279 | 0.309 | 0.269 |
| ETTh1 | 96 | MAE | 0.696 | 0.618 | 0.589 | **0.583** | 0.571 | 0.641 | 0.642 | 0.620 | 0.682 | 0.571 |
| | | MSE | 0.905 | 0.727 | 0.658 | 0.648 | 0.653 | 0.761 | 0.763 | 0.717 | 0.851 | **0.624** |
| | 192 | MAE | 0.820 | 0.617 | 0.647 | 0.583 | 0.599 | 0.619 | 0.619 | 0.598 | 0.658 | **0.568** |
| | | MSE | 1.165 | 0.725 | 0.817 | 0.640 | 0.719 | 0.687 | 1.619 | 0.665 | 0.788 | **0.598** |
| | 384 | MAE | 0.821 | 0.607 | 0.614 | 0.585 | 0.602 | 0.607 | 0.606 | 0.607 | 0.633 | **0.584** |
| | | MSE | 1.166 | 0.708 | 0.683 | 0.635 | 0.719 | 0.665 | 0.673 | 0.683 | 0.719 | **0.613** |
| | 768 | MAE | 0.820 | 0.625 | 0.749 | 0.641 | 0.636 | 0.616 | 0.623 | 0.624 | 0.641 | **0.599** |
| | | MSE | 1.163 | 0.734 | 1.029 | 0.733 | 0.811 | 0.676 | 0.706 | 0.721 | 0.733 | **0.649** |
| Weather | 96 | MAE | 0.408 | 0.409 | 0.427 | 0.498 | 0.543 | 0.413 | 0.394 | 0.388 | 0.439 | **0.336** |
| | | MSE | 0.336 | 0.348 | 0.357 | 0.440 | 0.545 | 0.364 | 0.337 | 0.332 | 0.392 | **0.267** |
| | 192 | MAE | 0.417 | 0.383 | 0.426 | 0.507 | 0.444 | 0.363 | 0.352 | 0.347 | 0.403 | **0.320** |
| | | MSE | 0.357 | 0.311 | 0.351 | 0.454 | 0.418 | 0.296 | 0.275 | 0.275 | 0.335 | **0.261** |
| | 384 | MAE | 0.452 | 0.381 | 0.405 | 0.517 | 0.654 | 0.359 | 0.345 | 0.338 | 0.405 | **0.334** |
| | | MSE | 0.401 | 0.314 | 0.329 | 0.477 | 0.698 | 0.292 | 0.269 | 0.265 | 0.333 | **0.256** |
| | 768 | MAE | 0.470 | 0.392 | 0.401 | 0.529 | 0.623 | 0.349 | 0.349 | 0.340 | 0.395 | 0.341 |
| | | MSE | 0.427 | 0.323 | 0.337 | 0.508 | 0.663 | 0.272 | 0.272 | **0.263** | 0.321 | 0.266 |
| Traffic | 96 | MAE | 0.676 | 0.483 | **0.428** | 0.439 | 0.516 | 0.443 | 0.438 | 0.440 | 0.504 | 0.442 |
| | | MSE | 1.240 | 0.905 | **0.759** | 0.708 | 0.907 | 0.821 | 0.819 | 0.812 | 0.874 | 0.786 |
| | 192 | MAE | 0.679 | 0.500 | 0.411 | 0.390 | 0.521 | 0.383 | 0.398 | 0.391 | 0.447 | **0.381** |
| | | MSE | 1.208 | 0.927 | 0.705 | 0.632 | 0.886 | 0.707 | 0.692 | 0.726 | 0.776 | **0.685** |
| | 384 | MAE | 0.678 | 0.503 | 0.399 | 0.393 | 0.486 | **0.379** | 0.420 | 0.385 | 0.444 | 0.383 |
| | | MSE | 1.197 | 0.953 | 0.696 | 0.648 | 0.795 | 0.702 | 0.755 | 0.716 | 0.772 | **0.700** |
| | 768 | MAE | 0.679 | 0.512 | 0.441 | 0.407 | – | 0.381 | **0.375** | 0.383 | 0.442 | 0.383 |
| | | MSE | 1.207 | 0.967 | 0.758 | 0.666 | – | 0.704 | **0.692** | 0.708 | 0.775 | 0.697 |

**Varying Missing Ratio.** Fig. 4 displays the performance of various methods under variable missing scenarios across the four datasets. As with time point missing, MAE is used as the evaluation metric, plotted against different missing ratios ($r$). Our method, S4M (indicated by the red line), consistently demonstrates competitive or superior performance across all datasets and missing ratios. In the Electricity dataset, S4M maintains one of the lowest MAEs, showing more stability compared to methods like GRU-D, which shows a sharp increase in error as the missing ratio grows. Similarly, in the ETTh1 and Weather datasets, S4M continues to outperform or match the best methods, particularly at higher missing ratios. For the Traffic dataset, while some methods perform comparably at lower missing ratios, S4M demonstrates robust resilience, with relatively low error even as the proportion of missing features increases. Overall, S4M shows strong generalization and consistent performance, effectively handling variable missing data scenarios across multiple datasets.

## 4.5 ABLATION STUDY

In the previous experiment, we investigated the effects of replacing the data inputs to the S4 backbone (blue columns in Tab. 1 and Tab. 2). To deepen the analysis, we conducted additional ablations on ATPM and the input stream of mask indications as shown in Tab. 3.

The results demonstrate the importance of incorporating the mask as the inputs to S4 backbone, as removing it consistently increases both MAE and MSE across various prediction horizons. Notably,

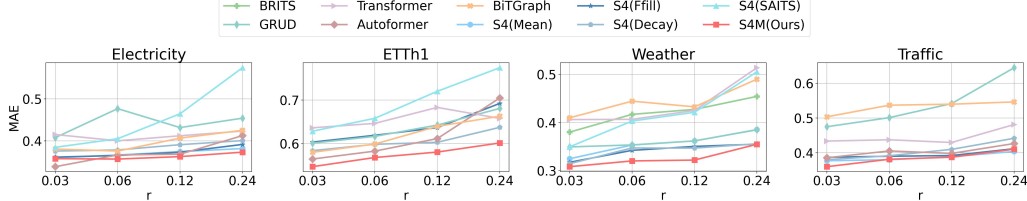

Figure 4: The performance of different methods on four datasets under variable missing scenario when the missing ratio $r$ varies from $0.03$ to $0.24$.

even when the error increases after removing masks appear numerically small in some entries, the overall predominantly positive red values reflect the model's enhanced stability and accuracy when handling missing data. This is particularly evident in the ETT and Weather datasets, where the presence of the mask significantly reduces errors, affirming the effectiveness of dual-inputs in `MDS-S4` to capture the complex dependencies inherent in multivariate time series with missing values.

The results also highlight the significance of ATPM. The model's performance improved significantly after incorporating ATPM, as both MSE and MAE increased across various settings when ATPM was removed, particularly on the Traffic and ETTh1 datasets. Additionally, ATPM demonstrated substantial improvements, especially with shorter lookback windows on the Electricity and Weather datasets, further emphasizing the improvements brought by ATPM.

Table 3: Results of ablation study for the mask and ATPM with blue values indicating a decrease in errors, while red values representing increase in errors.

| $\ell_L$ | Metric ↓ | Electricity S4M (Ours) | S4M (w/o mask) | S4M (w/o ATPM) | ETTh1 S4M (Ours) | S4M (w/o mask) | S4M (w/o ATPM) | Weather S4M (Ours) | S4M (w/o mask) | S4M (w/o ATPM) | Traffic S4M (Ours) | S4M (w/o mask) | S4M (w/o ATPM) |
|---|---|---|---|---|---|---|---|---|---|---|---|---|---|
| | | | | | | | Variable missing | | | | | | |
| 96 | MAE | 0.369 | +0.012 | +0.011 | 0.571 | -0.008 | +0.044 | 0.336 | +0.106 | +0.020 | 0.442 | +0.001 | +0.024 |
| | MSE | 0.282 | +0.010 | +0.010 | 0.624 | -0.008 | +0.091 | 0.267 | +0.520 | +0.206 | 0.786 | +0.039 | +0.125 |
| 192 | MAE | 0.357 | +0.004 | +0.010 | 0.568 | -0.013 | +0.045 | 0.320 | +0.061 | +0.600 | 0.381 | +0.003 | +0.030 |
| | MSE | 0.261 | +0.006 | +0.009 | 0.598 | -0.014 | +0.090 | 0.261 | +0.424 | +0.002 | 0.685 | +0.036 | +0.092 |
| 384 | MAE | 0.359 | +0.001 | +0.009 | 0.584 | +0.003 | +0.029 | 0.334 | +0.049 | +0.006 | 0.383 | +0.062 | +0.026 |
| | MSE | 0.264 | +0.002 | +0.009 | 0.613 | +0.008 | +0.064 | 0.256 | +0.444 | +0.008 | 0.700 | +0.092 | +0.065 |
| 768 | MAE | 0.362 | +0.004 | +0.020 | 0.599 | +0.012 | +0.028 | 0.341 | +0.043 | +0.016 | 0.383 | +0.000 | +0.026 |
| | MSE | 0.269 | +0.003 | +0.002 | 0.649 | +0.027 | +0.058 | 0.266 | +0.431 | +0.011 | 0.697 | +0.020 | +0.074 |
| | | | | | | | Time point missing | | | | | | |
| 96 | MAE | 0.372 | +0.014 | +0.025 | 0.571 | +0.003 | +0.049 | 0.313 | +0.035 | +0.021 | 0.428 | +0.003 | +0.045 |
| | MSE | 0.287 | +0.016 | +0.030 | 0.624 | +0.006 | +0.110 | 0.237 | +0.033 | +0.017 | 0.809 | +0.010 | +0.116 |
| 192 | MAE | 0.367 | +0.013 | +0.004 | 0.574 | -0.009 | +0.039 | 0.305 | +0.040 | +0.006 | 0.385 | +0.006 | +0.005 |
| | MSE | 0.274 | +0.012 | +0.004 | 0.593 | +0.022 | +0.110 | 0.225 | +0.041 | +0.001 | 0.687 | +0.034 | +0.023 |
| 384 | MAE | 0.370 | +0.002 | +0.014 | 0.571 | +0.012 | +0.057 | 0.306 | +0.040 | +0.012 | 0.385 | +0.000 | +0.013 |
| | MSE | 0.277 | +0.003 | +0.004 | 0.624 | +0.008 | +0.112 | 0.220 | +0.047 | +0.015 | 0.702 | -0.015 | +0.047 |
| 768 | MAE | 0.373 | -0.005 | +0.013 | 0.588 | +0.006 | +0.048 | 0.316 | +0.029 | +0.005 | 0.388 | -0.004 | +0.000 |
| | MSE | 0.282 | -0.003 | +0.016 | 0.647 | -0.001 | +0.079 | 0.232 | +0.037 | +0.004 | 0.699 | +0.011 | +0.024 |

## 5 CONCLUSION

In this paper, we present `S4M` for time series forecasting with missing values. `S4M` is an end-to-end framework that first uses a `ATPM` module to learn robust latent representation to account for missing values using rich historical data from a prototype bank, and then uses a missing-aware dual stream S4, `MDS-S4`, to directly model the mask of missing and the representation. The experimental results on four real-world benchmark datasets verify its superiority under various missing value scenarios. The ablation studies also show the importance of the masking mechanism in improving the model's robustness and accuracy. In the future, we would like to explore other S4-based architectures and missing types to make our proposed method more versatile.

## ACKNOWLEDGMENTS

Jing Peng and Qiong Zhang are supported by National Key R&D Program of China grant 2024YFA1015800. Xiaoxiao Li is supported by Natural Science and Engineering Research Council of Canada (NSERC), Canada CIFAR AI Chairs program, Canada Research Chair program, MITACS-CIFAR Catalyst Grant Program, the Digital Research Alliance of Canada.

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

## A    RELATED WORK

### A.1    TIME SERIES FORECASTING

Time series forecasting has seen major improvements thanks to both traditional statistical methods and modern deep learning models. The ARIMA model, for example, improves prediction accuracy by making non-stationary data more stable, which is a key method in time series analysis (Box & Jenkins, 1968). Recurrent Neural Networks (RNNs) have also become important tools in this field, providing a solid framework for modeling sequences and predicting time series, especially for capturing long-term patterns (Hewamalage et al., 2021). Improvements in RNN designs have led to different RNN-based approaches specifically made for forecasting (Rangapuram et al., 2018; Salinas et al., 2017; Lim et al., 2020). Attention-based models have gained attention because they can focus on key time steps, helping to capture long-term patterns that are critical for accurate forecasts (Qin et al., 2017; Shih et al., 2019). The encoder-decoder setup, in particular, has become a popular approach because of its strong forecasting ability. This has inspired various upgrades and new versions of the original Transformer model. One example is the Autoformer, which uses a new architecture with an Auto-Correlation mechanism, setting new standards for long-term forecasting accuracy (Wu et al., 2021). Similarly, the Pyraformer uses a pyramidal attention strategy to model different levels of data efficiently, boosting the accuracy of long-range time series predictions (Liu et al., 2022). The Scaleformer framework refines forecasts across different scales, leading to improved performance with little extra computation (Shabani et al., 2023). iTransformer introduces a novel approach by leveraging transformer-based architecture with adaptive self-attention mechanisms to capture temporal dependencies in time series forecasting (Liu et al., 2023). PatchTST applies a patch-based technique within a transformer framework to effectively capture both short- and long-term dependencies, improving forecasting accuracy across diverse time series tasks (Nie et al., 2022). CARD leverages a transformer architecture, focusing on aligning multiple temporal channels to capture dependencies effectively (Xue et al., 2023). Besides these advances, new models like the structured state space squence (S4) model combine the strengths of RNNs and CNNs, offering flexible solutions for a wide range of tasks, including generation, forecasting, and classification (Gu et al., 2021). S4 model combines the strengths of state-space models with modern deep learning architectures and can efficiently model long sequences.

### A.2    MISSING DATA IN TIME SERIES

In many real-world scenarios, datasets can be incomplete due to unforeseen events such as equipment failure or communication errors, making it crucial to address time series forecasting with missing data. GRU-D (Che et al., 2018) stands out as a classic method to manage missing data in recurrent models. Subsequent advances such as BRITS (Cao et al., 2018) have further refined the approach for LSTMs. The field has also seen the emergence of various imputation techniques, including M-RNN, GP-VAE, and SAITS, which prioritize the estimation of missing values to improve the precision of forecasting (Yoon et al., 2018; Fortuin et al., 2020; Du et al., 2023). Latent ODE (Rubanova et al., 2019), Neural ODE (Chen et al., 2018), CRUs (Schirmer et al., 2022), and GraFITi (Yalavarthi et al., 2024) each address missing values in time series through different mechanisms, with Latent ODE (Rubanova et al., 2019) and Neural ODE (Chen et al., 2018) learning continuous dynamics over time, CRUs (Schirmer et al., 2022) utilizing confidence regularization to improve imputation accuracy, and GraFITi (Yalavarthi et al., 2024) applying graph-based methods to capture temporal and spatial dependencies for missing data recovery. LGNet innovatively captures local and global temporal dynamics through a memory network (Tang et al., 2020). BiTGraph dexterously navigates temporal dependencies and spatial structures. By explicitly incorporating the challenge of missing values into its model architecture, BiTGraph aims to optimize the information flow and mitigate the adverse effects of data incompleteness (Chen et al., 2023).

## B    NOTATION TABLE

A summary of key notations used in the main paper is given in Tab. 4.

Table 4: Notations

| Notations | Description |
|---|---|
| $\boldsymbol{X}^{(L)}$ | look-back time series |
| $\boldsymbol{M}^{(L)}$ | mask: indicator for missing for look-back time series |
| $\boldsymbol{X}^{(H)}$ | horizon time series |
| $\boldsymbol{x}_t$ | raw value of the time series at time point $t$ |
| $\boldsymbol{o}_t$ | representation learning output at time point $t$ |
| $\boldsymbol{h}_t$ | S4 hidden state at time point $t$ |
| $\boldsymbol{y}_t$ | predicted value of S4 |
| $\boldsymbol{m}_t$ | mask at time point $t$ |
| $\boldsymbol{c}_j$ | the centroid of the $j$th cluster |
| $\ell_L$ | length of the look-back window |
| $\ell_H$ | length of the horizon window |
| $D$ | dimension of the time series |
| $R$ | encoder output dimension |
| $F$ | output channel of $\mathrm{ConvD}_1$ |
| $K_1$ | prototype bank parameter: maximum number of clusters |
| $K_2$ | prototype bank parameter: maximum number of elements within each cluster |
| $\tau_1, \tau_2$ | threshold for similarity in prototype bank writing |
| $\gamma$ | momentum coefficient |
| $\boldsymbol{\theta}$ | S4 model parameters |
| $t_0 : t_0 + \ell$ | $\{t_0, t_0 + 1, \ldots, t_0 + \ell\}$ |
| $E_p, E_q, E_m$ | encoder |

## C  ADDITIONAL DETAILS OF THE PROPOSED METHOD

In this section, we provide additional details of the proposed methods. We describe the procedure for local statistics extraction in Section C.1, the encoder design in the representation learning module in Section C.2, and the design of S4 blocks in Section C.3.

### C.1  LOCAL STATISTICS EXTRACTION

As the first step in dealing with missing values in time series, we extract useful local statistical features using contextual information from observed parts of the time series for missing values. Specifically, we denote $\boldsymbol{x}_{\min}, \boldsymbol{x}_{\max} \in \mathbb{R}^D$ respectively as the minimum and maximum of the observed value of $\boldsymbol{X}^{(L)}$. $\boldsymbol{\Delta}_{\min} \in \mathbb{R}^{\ell_L \times D}$, $\boldsymbol{\Delta}_{\max} \in \mathbb{R}^{\ell_L \times D}$ are the time gap between each entry of $\boldsymbol{X}$ with $\boldsymbol{x}_{\min}$, $\boldsymbol{x}_{\max}$. We use the combination of two exponential weights to extract local feature information from missing data. Specifically, we let

$$\boldsymbol{Z}^{(L)} = \boldsymbol{M}^{(L)} \boldsymbol{X}^{(L)} + (1 - \boldsymbol{M}^{(L)})(\boldsymbol{\Omega}_1' \boldsymbol{x}_{min} + \boldsymbol{\Omega}_2' \boldsymbol{x}_{max})$$

be the local statistics where

$$\boldsymbol{\Omega}_1 = \exp\{-\max(\boldsymbol{0}, \boldsymbol{W}_1 \boldsymbol{\Delta}_{\min} + \boldsymbol{b}_1)\}$$
$$\boldsymbol{\Omega}_2 = \exp\{-\max(\boldsymbol{0}, \boldsymbol{W}_2 \boldsymbol{\Delta}_{\max} + \boldsymbol{b}_2)\}$$
$$\boldsymbol{\Omega}_1' = \boldsymbol{\Omega}_1/(\boldsymbol{\Omega}_1 + \boldsymbol{\Omega}_2), \quad \boldsymbol{\Omega}_2' = \boldsymbol{\Omega}_2/(\boldsymbol{\Omega}_1 + \boldsymbol{\Omega}_2)$$

and $\boldsymbol{W}_1, \boldsymbol{W}_2, \boldsymbol{b}_1$ and $\boldsymbol{b}_2$ are the decay parameters. The local statistics $\boldsymbol{Z}$ are fed in the ATPM module to query from the prototype bank.

### C.2  ENCODER ARCHITECTURE

The architecture of the encoder $E_p$, $E_q$, and $E_m$ contains (1) a delay embedding layer, (2) a 2D-convolutional layer with ReLU activation, (3) a self-attention layer, and (4) a S4 layer. We describe these layers, respectively.

**Delay embedding** The delay embedding layer converts the original two-dimensional matrix $\boldsymbol{Z}^{(L)}$ (or $\boldsymbol{M}^{(L)}$ in $E_m$) into a third-order tensor. This technique involves recursively augmenting the

multivariate time series by unfolding the matrix along the temporal dimension. This process significantly enriches the local information at each time point by incorporating its historical time series data. Consequently, this enrichment facilitates the formalization and storage of various patterns.

**Convolution** We then incorporate a convolutional layer with a kernel size of $W$ in the temporal dimension and $D$ in the variable dimension to capture local temporal patterns and inter-variable dependencies. Subsequently, the output is passed through a Rectified Linear Unit (ReLU) layer. The ReLU layer's output is a matrix with dimensions $R \times T_c$ , where $R$ represents the number of filters in the convolutional layer and $T_c = L - W + 1$. Additionally, a dropout layer is applied subsequent to the ReLU layer to prevent overfitting.

**Attention** Subsequently, we implement an attention mechanism over the temporal dimension of the sequence, enabling the model to selectively emphasize salient information without changing the rank of tensor.

**S4 layer** The output from the attention layer is then fed into an S4 block. Unlike the layer of self-attention above, the S4 block was used to compress temporal information. Within this framework, we employ a S4 as an embedding tool, which serves to encapsulate the embedding of size $T_c \times R$ at each time point into a fixed-size representation vector of length $R$.

## C.3 Design of MDS-S4 blocks

Our second MDS-S4 module consists of one MDS-S4 block and multiple normal S4 blocks, each designed to process sequential data efficiently. The architecture begins with an MDS-S4 block. MDS-S4 is the core and initial layer of this block, which has dual inputs, the representation $o_t$ learned from `ATPM` and $\tilde{m}_t = E_m(m_t)$, both are fed into a dual-stream S4 block. To address the missing data problem in S4 models, we incorporate $\tilde{m}_t$ to distinguish the missing time points, enabling the model to treat them differently from the observed data (e.g., by referring to data in the prototype bank). At the same time, this structure ensures that the core properties of the S4 model are preserved. To this end, we seek a term that can flag missing values while preserving the HiPPO structure of S4. We find that integrating additional masking terms $M$, inspired by Che et al. (2018), to serve as a simple yet effective indicator for the model to recognize missing values. However, since the elements of $M$ take binary values (0 or 1), they are not naturally on the same scale as the other terms in equation 4. To address this, we design an encoder to transform the mask information to an appropriate scale. Incorporating this term still preserves the HiPPO structure of S4, thereby enriching the model with additional information while maintaining its core advantages. The output of the dual-stream S4 is then fed into a residual connection, coupled with layer normalization, to address gradient vanishing. Subsequently, a 1D convolutional layer with a kernel size of 1 and $F$ output channels is applied together with ReLU. Then, it comes another convolutional layer that reverts the output back to $R$ channels. Finally, a dropout layer is integrated to introduce regularization, which is crucial for preventing overfitting. The culmination of these operations completes a single MDS-S4 block within the architecture. We list these layers of the block in Tab. 5 for easy reference.

Table 5: Architecture of MDS-S4 block. For convolutional layer (Conv1D), we list parameters with sequence of input and output dimension, and kernel size.

| Layer | Details |
|---|---|
| 1 | MDS-S4 model or S4 model, Residual, LayerNorm |
| 2 | Conv1D($R$, $F$, 1), ReLU, Dropout |
| 3 | Conv2D($F$, $R$, 1), Dropout |

The following S4 blocks in MDS-S4 module have the same architecture with MDS-S4 block, except for the initial MDS-S4 model replaced with traditional S4 model. Begin with the MDS-S4 block, the output of one block is fed directly as input to the subsequent block.This iterative process allows for increasingly complex feature extraction and integration. The final output from the last block in the sequence represents `S4M`'s prediction.

# D EXPERIMENT DETAILS AND MORE RESULTS

## D.1 BASELINE METHODS

In this section, we describe the baseline methods that we compare with. The baselines include latest state-of-art methods and some classic methods. For models not specifically designed for missing data forecasting, we impute the missing observations with the mean value and conduct experiments on the imputed dataset.

- GRU-D: It is a time series model that extends the Gated Recurrent Unit (GRU) by incorporating decay mechanisms to handle missing data and capture temporal dependencies (Che et al., 2018).

- BRITS: It's a time series imputation model that integrates a Bidirectional Recurrent Neural Network (RNN) with a time decay mechanism to capture the relationships between missing values and observed data (Cao et al., 2018).

- Autoformer: It incorporates a seasonal decomposition mechanism that captures both long-term trends and short-term seasonal patterns. Autoformer also leverages auto-correlation to capture the dependencies between different time steps, allowing it to model the temporal relationships effectively (Wu et al., 2021).

- Transformer: It's a foundational sequential model that utilizes stacked self-attention blocks to effectively capture temporal dependencies in time series data (Vaswani et al., 2017).

- iTransformer: It introduces a novel methodology by integrating transformer-based architecture with adaptive self-attention mechanisms, enabling more efficient handling of complex temporal dependencies in time series forecasting tasks (Liu et al., 2023).

- CARD: It leverages a transformer architecture, focusing on aligning multiple temporal channels to capture dependencies effectively. Also, CARD incorporates a token blend module, which efficiently utilizes multi-scale knowledge to enhance the model's predictive power (Xue et al., 2023).

- CRUs: It introduces a unique method for handling missing or irregularly spaced data points, incorporating confidence-based regularization to improve the robustness and accuracy of time series forecasting models (Schirmer et al., 2022).

- GraFITi: A novel approach that models irregularly sampled time series data using graph-based techniques (Yalavarthi et al., 2024).

- BiTGraph: A state-of-the-art method that performs end-to-end prediction with biased temporal convolutional graph networks when missing data is present (Chen et al., 2023).

- S4 (Mean): Impute missing data using the global mean and employ S4 blocks as the backbone.

- S4 (Ffill): Impute missing data by forward filling with the latest observation, using S4 blocks as the backbone.

- S4 (Decay): Impute missing data by combining the global mean and the latest observation, with a decay factor controlling the weighting, and use S4 blocks as the backbone.

- S4 (SAITS): Fill missing entries with the state-of-the-art imputation method SAITS, using the imputed data as input for S4 blocks. SAITS is a time series forecasting method that employs a self-attention mechanism to capture long-term dependencies and trends, enabling more accurate imputation across various temporal patterns (Du et al., 2023).

We also provide detailed comparisons and computational cost analysis for above methods in Tab. 6. To measure the training and inference time, we conducted performance experiments using the electricity dataset, with a batch size of 16 and a hidden size of 512. The maximum memory usage, along with the training and inference times, were recorded for a single epoch.

We observe that the ODE-based method incurs an extraordinarily high computational cost compared to other approaches. Notably, S4M demonstrates significantly lower memory usage compared to other state-of-the-art transformer-based methods, particularly iTransformer and CARD. Additionally, S4-based methods generally exhibit shorter training times compared to recurrent methods

designed for forecasting with missing observations, such as BRITS and GRU-D. These results reinforce our decision to focus on S4-based architectures due to their superior efficiency.

Table 6: Computation Cost for Different Methods ('OOM' refers to "Out-of-Memory").

| Method | GRU-D | CRUs | GraFITi | S4(Mean) | S4(Decay) | S4M(Ours) |
|---|---|---|---|---|---|---|
| Memory(MB) | 445.70 | 32146.06 | OOM | 764.75 | 887.28 | 1071.39 |
| Training Time(s) | 244.26 | 64990.81 | OOM | 84.71 | 90.07 | 110.65 |
| Inference Time(s) | 32.64 | 1524.16 | OOM | 12.16 | 13.23 | 14.55 |
| Method | Autoformer | BiTGraph | Transformer | iTransformer | CARD | BRITS |
| Memory(MB) | 1106.41 | 2564.35 | 777.69 | 2107.67 | 7517.68 | 557.68 |
| Training Time(s) | 95.98 | 267.74 | 78.00 | 74.16 | 274.29 | 486.68 |
| Inference Time(s) | 17.88 | 24.88 | 10.35 | 12.31 | 34.40 | 57.19 |

## D.2 DATASET DETAILS

In Tab. 7, we present the number of variables (Variables), the total length of the time series (Time steps), and the frequency that observations are made (Granularity). We select these four datasets because they exhibit significant variation in size, number of variables, and the presence or absence of seasonality.

Table 7: Dataset statistics.

| Data | Variables | Time steps | Granularity |
|---|---|---|---|
| Electricity | 321 | 26,304 | 1 hour |
| ETTh1 | 7 | 17,420 | 15 min |
| Weather | 21 | 52,696 | 10 min |
| Traffic | 862 | 17,544 | 1 hour |

For all datasets in our experiment, we consider two different missing scenarios: *time point missing* and *variable missing*, which is illustrated in Fig. 5. Under the time point missing scenario, we first

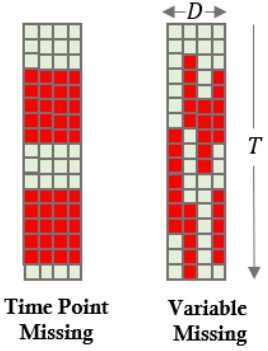

Figure 5: Illustration of block missing patterns: Time Point Missing (Left) and Variable Missing (Right). Each column represents a variable in the time series, and each row corresponds to observations at a specific time point. Red blocks indicate missing observations, while white blocks represent observed data. Missing values are consecutive in both patterns. For time point missing, all variables are missing at a given time point. For variable missing, some variables may remain observed at the same time point.

randomly select a ratio $r$ of time points, and for each selected time point, we remove its following consecutive time points of length 5 and eliminate all variables at those time points. In the variable missing scenario, we perform the same procedure independently for each variable. When generating the missing data, the missing ratio $r$ ranges from 0.03, 0.06, 0.12, to 0.24. Due to the design of the

consecutive missing points, the overall missing ratio (the percentage of missing entries in the times series matrix) is higher than $r$, and we report these values in Tab. 8 under different values of $r$.

Table 8: Overall missing ratio statistics.

| Missing pattern | $r$ | 0.030 | 0.060 | 0.120 | 0.240 |
|---|---|---|---|---|---|
| Time point missing | Electricity | 0.139 | 0.260 | 0.450 | 0.694 |
| | ETTh1 | 0.122 | 0.231 | 0.399 | 0.616 |
| | Traffic | 0.139 | 0.256 | 0.447 | 0.705 |
| | Weather | 0.132 | 0.247 | 0.432 | 0.667 |
| Variable missing | Electricity | 0.139 | 0.258 | 0.450 | 0.696 |
| | ETTh1 | 0.122 | 0.228 | 0.395 | 0.613 |
| | Traffic | 0.139 | 0.259 | 0.451 | 0.698 |
| | Weather | 0.133 | 0.258 | 0.431 | 0.667 |

### D.3 HYPER-PARAMETER DETAILS

The learning rates are set to $0.01$ for the Electricity and Traffic datasets, $0.005$ for the ETTh1 dataset, and $0.001$ for the Weather dataset. The dimensions of the hidden layers are set to $512$ for the Electricity and Traffic datasets, and $256$ for the ETTh1 and Weather datasets. The number of basic blocks or layers is selected from $\{2, 4, 8\}$. The batch size set for all experiments are $16$. We use the Adam optimizer and implement an early stopping strategy across all experiments. Other hyperparameters for both the proposed method and baseline methods are adjusted based on their performance on the validation set. The performance of different methods is evaluated using Mean Squared Error (MSE) and Mean Absolute Error (MAE). For both metrics, lower values indicate better performance.

### D.4 SENSITIVITY ANALYSIS

In this section, we evaluate the sensitivity of our method with respect to the size of queue $K_1$, $K_2$, threshold $\tau_1$, $\tau_2$, the dimension $R$ of encoder output, the size of the short period retrieve window $s$, and the number of memory centroids $K$ we choose. All of the experiments are done under the time point missing scenario with $r = 0.06$, look-back window $H = 96$, which is a representative scenario to make analysis on.

#### D.4.1 BANK INITIALIZATION

In our experiment, we find the performance is insensitive to the initial cluster configuration, as the clusters are updated continuously throughout the training process. To provide evidence, we present the experimental results on four datasets using different cluster numbers for initialization, as shown below in Tab. 9. In practice, we recommend using $3$ to $5$ clusters for initialization or determining the optimal number of clusters based on the within-cluster sum of squares.

Table 9: Performance of S4M (Ours) with different number of clusters for initialization with other parameters fixed.

| number of clusters | | 1 | 2 | 3 | 4 | 8 | 12 | 16 |
|---|---|---|---|---|---|---|---|---|
| Electricity | MAE | 0.415 | 0.415 | 0.415 | 0.415 | 0.415 | 0.415 | 0.415 |
| | MSE | 0.356 | 0.356 | 0.356 | 0.356 | 0.357 | 0.358 | 0.356 |
| ETTh1 | MAE | 0.647 | 0.647 | 0.648 | 0.648 | 0.647 | 0.648 | 0.65 |
| | MSE | 0.768 | 0.767 | 0.77 | 0.77 | 0.767 | 0.767 | 0.773 |
| Weather | MAE | 0.386 | 0.39 | 0.387 | 0.385 | 0.388 | 0.388 | 0.385 |
| | MSE | 0.307 | 0.31 | 0.308 | 0.306 | 0.31 | 0.31 | 0.307 |
| Traffic | MAE | 0.510 | 0.505 | 0.504 | 0.515 | 0.509 | 0.513 | 0.509 |
| | MSE | 0.966 | 0.954 | 0.944 | 0.999 | 0.974 | 0.992 | 0.985 |

D.4.2 ANALYSIS OF $K_1$ AND $K_2$

We fix $\tau_1 = 0.95$, $\tau_2 = 0.6$, $R = 256$, and $s = 32$. $K_1$ represents the size of the maximum centroid, which governs the storage of prototype clusters. Choosing an appropriate value for $K_1$ allows the bank to effectively filter out outdated representations, especially in cases with a large number of patterns in the original time series. Tab. 10 indicates that a suitable value for $K_1$ is below 50.

For the analysis of $K_2$, we set $K_1 = 30$. $K_2$ controls the size of each cluster in the prototype bank. A smaller $K_2$ allows the bank to store only newly generated representations, ensuring that it remains aligned with the model's updates. Tab. 11 shows the performance changes across different values of $K_2$, suggesting that a relatively smaller value is more beneficial. We do not include results for ETTh1 because its shorter time series length and variable dimensions result in a significantly smaller pattern size, which does not require a constraint on the number of clusters.

Table 10: Performance of S4M (Ours) when $K_1 = 5, 19, 30, 50$, and $100$ with other parameters fixed.

| Data | Metric ↓ | 5 | 10 | 30 | 50 | 100 |
|---|---|---|---|---|---|---|
| Electricity | MSE | **0.372** | 0.377 | 0.377 | 0.376 | 0.376 |
| | MAE | **0.287** | 0.293 | 0.293 | 0.293 | 0.290 |
| Weather | MSE | 0.347 | **0.345** | 0.345 | 0.347 | 0.347 |
| | MAE | 0.270 | **0.267** | 0.267 | 0.268 | 0.268 |
| Traffic | MSE | 0.442 | 0.438 | 0.437 | **0.436** | 0.439 |
| | MSE | 0.863 | 0.823 | 0.819 | **0.817** | 0.830 |

Table 11: Performance of S4M (Ours) when $K_2 = 3, 5, 10, 20, 50$, and $100$ with other parameters fixed.

| Data | Metric ↓ | 3 | 5 | 10 | 20 | 50 | 100 |
|---|---|---|---|---|---|---|---|
| Electricity | MAE | 0.393 | **0.377** | 0.393 | 0.398 | 0.393 | 0.394 |
| | MSE | 0.313 | **0.299** | 0.312 | 0.319 | 0.312 | 0.313 |
| ETTh1 | MAE | 0.606 | 0.610 | 0.607 | 0.603 | 0.605 | **0.601** |
| | MSE | 0.695 | 0.700 | 0.694 | 0.655 | 0.659 | **0.655** |
| Weather | MAE | 0.347 | 0.345 | **0.343** | 0.346 | 0.346 | 0.347 |
| | MSE | 0.268 | **0.267** | 0.268 | 0.268 | 0.268 | 0.268 |
| Traffic | MAE | 0.438 | **0.436** | 0.438 | 0.437 | 0.438 | 0.438 |
| | MSE | **0.815** | 0.818 | 0.827 | 0.828 | 0.830 | 0.822 |

D.4.3 ANALYSIS OF $R$ AND $s$

In this section, we performe sensitivity analysis when the dimension of the encoder $R$ and the short period window size $s$ varies. We set $K_1 = 30$, $K_2 = 50$, $\tau_1 = 0.95$, and $\tau_2 = 0.6$. For the analysis of $R$, we fix $s = 16$ and vary the values of $R$ from 16 to 1024. Similarly, for the analysis of $s$, we set $R = 256$ and vary the values of $s$ from 8 to 48. Tab. 12 shows that $R$ significantly affects performance, with values larger than 128 benefiting the model. Tab. 13 shows that increasing $s$ generally improves performance.

D.4.4 ANALYSIS OF $\tau_1$ AND $\tau_2$

We set $K_1 = 30$, $K_2 = 50$, $s = 16$, and $R = 256$, and then vary the values of $\tau_1$ and $\tau_2$ across four datasets to observe how changes in these thresholds affect model performance in Tab. 14 – Tab. 17. Overall, the forecasting performance is less sensitive to $\tau_1$ and $\tau_2$ compared to other hyperparameters we previously analyzed. Specifically, model performance on ETTh1 and Traffic is more sensitive to these threshold values than on the other two datasets. ETTh1 achieves its best performance when $\tau_1 \le 0.95$ and $\tau_2 \le 0.9$, while Traffic performs optimally at $\tau_1 = 0.9$ and $\tau_2 = 0.5$. Electricity and Weather exhibit similar patterns, with slight performance improvements when $\tau_1 = 0.975$ and $\tau_2 = 0.5$.

Table 12: Performance of S4M (Ours) when $R$ ranging from 16 to 1024 with other parameters fixed.

| Dataset | Metric ↓ | 16 | 32 | 64 | 128 | 256 | 512 | 1024 |
|---|---|---|---|---|---|---|---|---|
| Electricity | MAE | 0.409 | 0.400 | 0.406 | 0.388 | 0.376 | 0.379 | **0.375** |
| | MSE | 0.358 | 0.335 | 0.339 | 0.308 | **0.279** | 0.295 | 0.292 |
| ETTh1 | MAE | 0.480 | 0.438 | 0.465 | 0.444 | 0.418 | 0.418 | **0.400** |
| | MSE | 0.895 | 0.826 | 0.846 | 0.855 | 0.363 | 0.351 | **0.332** |
| Weather | MAE | 0.585 | 0.591 | 0.603 | 0.571 | **0.571** | 0.610 | 0.609 |
| | MSE | 0.654 | 0.656 | 0.680 | 0.624 | **0.621** | 0.699 | 0.690 |
| Traffic | MAE | 0.329 | 0.320 | 0.329 | **0.315** | 0.352 | 0.317 | 0.349 |
| | MSE | 0.257 | 0.246 | 0.255 | **0.243** | 0.277 | 0.244 | 0.274 |

Table 13: Performance of S4M (Ours) when $s$ varies with other parameters fixed.

| Dataset | $s$ | 8 | 16 | 32 | 48 |
|---|---|---|---|---|---|
| Electricity | MAE | 0.379 | 0.379 | 0.378 | **0.378** |
| | MSE | 0.297 | 0.296 | 0.295 | **0.293** |
| ETTh1 | MAE | 0.596 | **0.570** | 0.584 | 0.582 |
| | MSE | 0.660 | **0.624** | 0.656 | 0.649 |
| Weather | MAE | 0.345 | 0.350 | 0.351 | **0.332** |
| | MSE | 0.267 | 0.275 | 0.279 | **0.253** |
| Traffic | MAE | 0.453 | 0.438 | 0.443 | **0.436** |
| | MSE | 0.847 | 0.826 | 0.853 | **0.786** |

### D.4.5 MORE DISCUSSION ON HYPERPARAMETERS $K_1$, $K_2$, $\tau_1$, AND $\tau_2$.

Based on our results, we recommend setting $K_2$ between 5 and 10, as this range is both effective and relatively insensitive to performance variations. The suggested ranges for $\tau_1$ and $\tau_2$ are $[0.3, 0.6]$ and $[0.8, 1.0)$, respectively, with both parameters selectable using the validation set. Most experiments are robust to choices of $K_1 = 30$ or $K_1 = 50$. If a dataset contains a very large number of clusters, as determined by a preliminary experiment with a higher $K_1$ value, the number of clusters can be adjusted accordingly.

Table 14: Performance under different values of $\tau_1$ and $\tau_2$ on Electricity. Entries with '–' mean the experiment is not meaningful in our setting because we set $\tau_1 \geq \tau_2$.

| $\tau_1 \backslash \tau_2$ | Metric ↓ | 0.050 | 0.100 | 0.300 | 0.500 | 0.700 | 0.900 |
|---|---|---|---|---|---|---|---|
| 0.500 | MAE | 0.393 | 0.392 | 0.392 | 0.393 | – | – |
| | MSE | 0.312 | 0.311 | 0.311 | 0.311 | – | – |
| 0.700 | MAE | 0.393 | 0.393 | 0.393 | 0.393 | 0.393 | – |
| | MSE | 0.311 | 0.312 | 0.312 | 0.312 | 0.312 | – |
| 0.900 | MAE | 0.392 | 0.391 | 0.392 | 0.392 | 0.393 | 0.393 |
| | MSE | 0.311 | 0.310 | 0.311 | 0.311 | 0.312 | 0.311 |
| 0.950 | MAE | 0.395 | 0.395 | 0.394 | 0.393 | 0.393 | 0.393 |
| | MSE | 0.316 | 0.315 | 0.315 | 0.313 | 0.314 | 0.312 |
| 0.975 | MAE | 0.392 | 0.395 | 0.395 | 0.394 | 0.393 | 0.393 |
| | MSE | 0.311 | 0.315 | 0.316 | 0.314 | 0.314 | 0.312 |

Table 15: Performance under different values of $\tau_1$ and $\tau_2$ on Weather. Entries with '–' mean the experiment is not meaningful in our setting because we set $\tau_1 \geq \tau_2$.

| $\tau_1 \backslash \tau_2$ | Metric ↓ | 0.050 | 0.100 | 0.300 | 0.500 | 0.700 | 0.900 |
|---|---|---|---|---|---|---|---|
| 0.500 | MAE | 0.342 | 0.342 | 0.342 | 0.342 | – | – |
| | MSE | 0.269 | 0.269 | 0.269 | 0.268 | – | – |
| 0.700 | MAE | 0.343 | 0.342 | 0.343 | 0.343 | 0.342 | – |
| | MSE | 0.270 | 0.270 | 0.270 | 0.270 | 0.269 | – |
| 0.900 | MAE | 0.343 | 0.343 | 0.343 | 0.343 | 0.343 | 0.343 |
| | MSE | 0.271 | 0.271 | 0.271 | 0.271 | 0.270 | 0.270 |
| 0.950 | MAE | 0.340 | 0.341 | 0.341 | 0.340 | 0.341 | 0.344 |
| | MSE | 0.267 | 0.268 | 0.268 | 0.267 | 0.268 | 0.270 |
| 0.975 | MAE | 0.339 | 0.339 | 0.339 | 0.340 | 0.342 | 0.345 |
| | MSE | 0.266 | 0.266 | 0.266 | 0.267 | 0.269 | 0.270 |

Table 16: Performance under different values of $\tau_1$ and $\tau_2$ on ETTh1. Entries with '–' mean the experiment is not meaningful in our setting because we set $\tau_1 \geq \tau_2$.

| $\tau_1 \backslash \tau_2$ | Metric ↓ | 0.050 | 0.100 | 0.300 | 0.500 | 0.700 | 0.900 |
|---|---|---|---|---|---|---|---|
| 0.500 | MAE | 0.591 | 0.591 | 0.591 | 0.591 | – | – |
| | MSE | 0.662 | 0.662 | 0.663 | 0.663 | – | – |
| 0.700 | MAE | 0.591 | 0.591 | 0.591 | 0.591 | 0.591 | – |
| | MSE | 0.663 | 0.662 | 0.663 | 0.662 | 662 | – |
| 0.900 | MAE | 0.591 | 0.591 | 0.591 | 0.591 | 0.591 | 0.591 |
| | MSE | 0.659 | 0.659 | 0.659 | 0.659 | 0.659 | – |
| 0.950 | MAE | 0.591 | 0.591 | 0.591 | 0.591 | 0.591 | 0.597 |
| | MSE | 0.651 | 0.651 | 0.651 | 0.651 | 0.651 | 0.671 |
| 0.975 | MAE | 0.580 | 0.582 | 0.583 | 0.582 | 0.586 | 0.599 |
| | MSE | 0.643 | 0.647 | 0.647 | 0.644 | 0.650 | 0.680 |

Table 17: Performance under different values of $\tau_1$ and $\tau_2$ on Traffic. Entries with '–' mean the experiment is not meaningful in our setting because we set $\tau_1 \geq \tau_2$.

| $\tau_1 \backslash \tau_2$ | Metric ↓ | 0.050 | 0.100 | 0.300 | 0.500 | 0.700 | 0.900 |
|---|---|---|---|---|---|---|---|
| 0.500 | MAE | 0.444 | 0.442 | 0.445 | 0.442 | – | – |
| | MSE | 0.870 | 0.855 | 0.870 | 0.856 | – | – |
| 0.700 | MAE | 0.442 | 0.439 | 0.441 | 0.440 | 0.441 | – |
| | MSE | 0.854 | 0.836 | 0.848 | 0.833 | 0.857 | – |
| 0.900 | MAE | 0.441 | 0.441 | 0.440 | 0.438 | 0.440 | 0.441 |
| | MSE | 0.851 | 0.840 | 0.849 | 0.808 | 0.852 | 0.857 |
| 0.950 | MAE | 0.439 | 0.439 | 0.446 | 0.440 | 0.444 | 0.439 |
| | MSE | 0.837 | 0.834 | 0.869 | 0.842 | 0.859 | 0.838 |
| 0.975 | MAE | 0.445 | 0.443 | 0.442 | 0.438 | 0.445 | 0.439 |
| | MSE | 0.865 | 0.848 | 0.857 | 0.825 | 0.872 | 0.851 |

### D.5 EXPERIMENT RESULT ON DATASET WITH NO MISSING VALUE

If there is no missing data, as expected in Tab. 18, our method does not show a significant advantage over the other methods but still maintains a very competitive performance. For this experiment, we report the results using a horizon window of size 96 and a lookback window of size 96, with no missing values in the original dataset.

Table 18: Comparison of forecasting performance of S4M (Ours) and baselines with $\ell_H = 96$, $\ell_L = 96$ on four datasets with no missing value.

| Dataset | Metric ↓ | BRITS | GRU-D | Transformer | Autoformer | S4 | BiTGraph | S4M(Ours) |
|---|---|---|---|---|---|---|---|---|
| Electricity | MAE | 0.398 | 0.413 | 0.411 | 0.352 | 0.386 | 0.348 | 0.383 |
| | MSE | 0.318 | 0.332 | 0.321 | 0.242 | 0.301 | 0.254 | 0.295 |
| ETTh1 | MAE | 0.676 | 0.571 | 0.604 | 0.556 | 0.538 | 0.530 | 0.538 |
| | MSE | 0.867 | 0.636 | 0.677 | 0.588 | 0.560 | 0.571 | 0.560 |
| Weather | MAE | 0.373 | 0.370 | 0.383 | 0.306 | 0.363 | 0.504 | 0.332 |
| | MSE | 0.291 | 0.301 | 0.298 | 0.235 | 0.301 | 0.494 | 0.259 |
| Traffic | MAE | 0.428 | 0.446 | 0.405 | 0.454 | 0.425 | 0.504 | 0.414 |
| | MSE | 0.770 | 0.840 | 0.707 | 0.705 | 0.425 | 0.879 | 0.761 |

### D.6 ADDITIONAL EXPERIMENT RESULTS

In the main text of the manuscript, we include the comparison of S4M with different baselines under the missing ratio $r = 0.06$. In this section, we provide the complete additional results in Tab. 19 to Tab. 24 when $r = 0.03$, $r = 0.12$, and $r = 0.24$. Similar to the $r = 0.06$ case, Our proposed S4M consistently achieves the best or second-best performance across most settings, demonstrating its robustness in handling missing data.

Table 19: Comparison of forecasting performance of S4M (Ours) and baselines on four datasets with various look-back window length under time point missing scenario when missing ratio $r = 0.03$.

| Data | $\ell_L$ | Metric ↓ | BRITS | GRU-D | Trans. | Auto. | BiTGraph | S4 (Mean) | S4 (Ffill) | S4 (Decay) | S4 (SAITS) | S4M (Ours) |
|---|---|---|---|---|---|---|---|---|---|---|---|---|
| Electricity | 96 | MAE | 0.606 | 0.419 | 0.413 | 0.374 | 0.390 | 0.395 | 0.409 | 0.397 | 0.409 | **0.370** |
| | | MSE | 0.579 | 0.338 | 0.329 | 0.272 | 0.300 | 0.316 | 0.333 | 0.312 | 0.334 | **0.281** |
| | 192 | MAE | 0.616 | 0.421 | 0.409 | **0.348** | 0.380 | 0.380 | 0.383 | 0.372 | 0.395 | 0.369 |
| | | MSE | 0.595 | 0.342 | 0.318 | **0.240** | 0.280 | 0.288 | 0.289 | 0.274 | 0.303 | 0.272 |
| | 384 | MAE | 0.627 | 0.420 | 0.420 | **0.346** | 0.366 | 0.377 | 0.384 | 0.378 | 0.392 | 0.371 |
| | | MSE | 0.619 | 0.339 | 0.333 | **0.240** | 0.264 | 0.285 | 0.289 | 0.278 | 0.300 | 0.273 |
| | 768 | MAE | 0.635 | 0.419 | 0.409 | **0.353** | 0.391 | 0.378 | 0.382 | 0.375 | 0.392 | 0.372 |
| | | MSE | 0.632 | 0.338 | 0.324 | **0.251** | 0.289 | 0.286 | 0.286 | 0.276 | 0.299 | 0.273 |
| ETTh1 | 96 | MAE | 0.696 | 0.624 | 0.681 | 0.624 | **0.528** | 0.618 | 0.625 | 0.603 | 0.632 | 0.565 |
| | | MSE | 0.917 | 0.734 | 0.885 | 0.752 | **0.556** | 0.721 | 0.732 | 0.689 | 0.757 | 0.603 |
| | 192 | MAE | 0.731 | 0.629 | 0.669 | 0.619 | 0.607 | 0.596 | 0.599 | 0.588 | 0.619 | **0.555** |
| | | MSE | 0.971 | 0.742 | 0.883 | 0.739 | 0.736 | 0.661 | 0.663 | 0.650 | 0.710 | **0.566** |
| | 384 | MAE | 0.745 | 0.625 | 0.698 | 0.625 | **0.545** | 0.597 | 0.602 | 0.599 | 0.616 | **0.557** |
| | | MSE | 1.010 | 0.734 | 0.933 | 0.746 | 0.599 | 0.663 | 0.669 | 0.669 | 0.697 | **0.586** |
| | 768 | MAE | 0.781 | 0.646 | 1.156 | 0.651 | 0.623 | 0.616 | 0.618 | 0.614 | 0.623 | **0.580** |
| | | MSE | 1.061 | 0.780 | 1.157 | 0.768 | 0.760 | 0.695 | 0.696 | 0.700 | 0.711 | **0.624** |
| Weather | 96 | MAE | 0.408 | 0.402 | 0.436 | 0.400 | 0.534 | 0.372 | 0.366 | 0.388 | 0.424 | **0.345** |
| | | MSE | 0.327 | 0.336 | 0.365 | 0.327 | 0.531 | 0.305 | 0.298 | 0.331 | 0.375 | **0.281** |
| | 192 | MAE | 0.378 | 0.378 | 0.420 | 0.412 | 0.433 | 0.350 | 0.337 | 0.345 | 0.374 | **0.315** |
| | | MSE | 0.303 | 0.303 | 0.351 | 0.342 | 0.401 | 0.268 | 0.255 | 0.271 | 0.297 | **0.246** |
| | 384 | MAE | 0.375 | 0.375 | 0.414 | 0.421 | 0.653 | 0.338 | **0.326** | 0.337 | 0.373 | 0.333 |
| | | MSE | 0.305 | 0.305 | 0.345 | 0.363 | 0.694 | 0.263 | **0.251** | 0.261 | 0.294 | 0.256 |
| | 768 | MAE | 0.385 | 0.385 | 0.394 | 0.448 | 0.618 | 0.351 | 0.340 | 0.333 | 0.370 | 0.336 |
| | | MSE | **0.314** | **0.314** | 0.329 | 0.407 | 0.655 | 0.273 | 0.261 | **0.255** | 0.291 | 0.259 |
| Traffic | 96 | MAE | 0.677 | 0.504 | 0.449 | 0.471 | 0.516 | 0.455 | 0.444 | 0.433 | 0.455 | **0.420** |
| | | MSE | 1.198 | 0.923 | 0.788 | **0.767** | 0.915 | 0.837 | 0.822 | 0.811 | 0.837 | 0.849 |
| | 192 | MAE | 0.681 | 0.501 | 0.437 | 0.405 | 0.537 | 0.404 | 0.399 | 0.391 | 0.413 | **0.381** |
| | | MSE | 1.225 | 0.927 | 0.754 | **0.648** | 0.944 | 0.698 | 0.710 | 0.710 | 0.711 | 0.697 |
| | 384 | MAE | 0.680 | 0.507 | 0.417 | 0.390 | 0.527 | 0.401 | 0.392 | 0.385 | 0.406 | **0.380** |
| | | MSE | 1.216 | 0.940 | 0.715 | **0.632** | 0.908 | 0.695 | 0.700 | 0.703 | 0.701 | 0.689 |
| | 768 | MAE | 0.680 | 0.507 | 0.456 | 0.435 | – | 0.391 | 0.389 | 0.388 | 0.396 | **0.380** |
| | | MSE | 1.223 | 0.882 | 0.772 | 0.715 | – | 0.693 | 0.707 | 0.731 | 0.694 | 0.696 |

Table 20: Comparison of forecasting performance of S4M (Ours) and baselines on four datasets with various look-back window length under variable missing scenario when missing ratio $r = 0.03$.

| Data | $\ell_L$ | Metric ↓ | BRITS | GRU-D | Trans. | Auto. | BiTGraph | S4 (Mean) | S4 (Ffill) | S4 (Decay) | S4 (SAITS) | S4M (Ours) |
|---|---|---|---|---|---|---|---|---|---|---|---|---|
| Electricity | 96 | MAE | 0.415 | 0.424 | 0.401 | **0.356** | 0.373 | 0.384 | 0.386 | 0.389 | 0.403 | 0.379 |
| | | MSE | 0.339 | 0.351 | 0.312 | **0.250** | 0.280 | 0.301 | 0.303 | 0.305 | 0.321 | 0.290 |
| | 192 | MAE | 0.423 | 0.408 | 0.415 | **0.338** | 0.381 | 0.370 | 0.371 | 0.376 | 0.385 | 0.358 |
| | | MSE | 0.349 | 0.327 | 0.326 | **0.226** | 0.281 | 0.275 | 0.281 | 0.280 | 0.289 | 0.260 |
| | 384 | MAE | 0.439 | 0.429 | 0.409 | **0.359** | 0.380 | 0.364 | 0.368 | 0.374 | 0.384 | 0.362 |
| | | MSE | 0.368 | 0.361 | 0.316 | **0.251** | 0.279 | 0.268 | 0.274 | 0.278 | 0.291 | 0.265 |
| | 768 | MAE | 0.437 | 0.445 | 0.406 | **0.358** | 0.376 | 0.364 | 0.373 | 0.373 | 0.388 | 0.362 |
| | | MSE | 0.370 | 0.378 | 0.323 | **0.253** | 0.274 | 0.267 | 0.284 | 0.277 | 0.294 | 0.265 |
| ETTh1 | 96 | MAE | 0.691 | 0.599 | 0.607 | 0.593 | **0.544** | 0.645 | 0.623 | 0.606 | 0.644 | 0.560 |
| | | MSE | 0.892 | 0.678 | 0.683 | 0.681 | 0.600 | 0.757 | 0.714 | 0.686 | 0.786 | **0.598** |
| | 192 | MAE | 0.725 | 0.601 | 0.686 | 0.564 | 0.580 | 0.605 | 0.603 | 0.584 | 0.628 | **0.547** |
| | | MSE | 0.943 | 0.679 | 0.890 | 0.614 | 0.682 | 0.605 | 0.668 | 0.631 | 0.732 | **0.574** |
| | 384 | MAE | 0.738 | 0.600 | 0.603 | 0.596 | 0.581 | 0.600 | 0.591 | 0.601 | 0.627 | **0.556** |
| | | MSE | 0.982 | 0.680 | 0.672 | 0.673 | 0.680 | 0.661 | 0.636 | 0.676 | 0.730 | **0.593** |
| | 768 | MAE | 0.771 | 0.607 | 0.759 | 0.619 | 0.619 | 0.600 | 0.606 | 0.612 | 0.642 | **0.569** |
| | | MSE | 1.024 | 0.689 | 0.967 | 0.672 | 0.744 | 0.661 | 0.665 | 0.690 | 0.766 | **0.599** |
| Weather | 96 | MAE | 0.375 | 0.373 | 0.384 | 0.377 | 0.511 | 0.375 | 0.362 | 0.360 | 0.388 | **0.340** |
| | | MSE | 0.298 | 0.308 | 0.306 | 0.296 | 0.505 | 0.319 | 0.302 | 0.300 | 0.329 | **0.272** |
| | 192 | MAE | 0.380 | 0.349 | 0.406 | 0.388 | 0.410 | 0.325 | 0.317 | 0.314 | 0.349 | **0.308** |
| | | MSE | 0.317 | 0.278 | 0.332 | 0.311 | 0.374 | 0.249 | 0.237 | 0.239 | 0.270 | **0.227** |
| | 384 | MAE | 0.417 | 0.357 | 0.369 | 0.403 | 0.626 | 0.324 | 0.315 | 0.306 | 0.347 | **0.302** |
| | | MSE | 0.358 | 0.287 | 0.288 | 0.338 | 0.662 | 0.247 | 0.234 | 0.230 | 0.266 | **0.222** |
| | 768 | MAE | 0.437 | 0.362 | 0.372 | 0.421 | 0.603 | 0.325 | 0.317 | 0.306 | 0.342 | **0.300** |
| | | MSE | 0.387 | 0.294 | 0.306 | 0.374 | 0.635 | 0.246 | 0.235 | 0.225 | 0.342 | **0.220** |
| Traffic | 96 | MAE | 0.680 | 0.459 | 0.439 | 0.452 | 0.510 | **0.427** | 0.431 | 0.437 | 0.479 | 0.431 |
| | | MSE | 1.211 | 0.845 | 0.756 | **0.746** | 0.894 | 0.779 | 0.805 | 0.791 | 0.847 | 0.798 |
| | 192 | MAE | 0.669 | 0.475 | 0.434 | 0.386 | 0.504 | 0.377 | 0.387 | 0.380 | 0.419 | **0.360** |
| | | MSE | 1.181 | 0.873 | 0.741 | **0.619** | 0.839 | 0.694 | 0.727 | 0.694 | 0.738 | 0.598 |
| | 384 | MAE | 0.669 | 0.467 | 0.397 | 0.389 | 0.482 | 0.373 | 0.383 | 0.375 | 0.413 | **0.372** |
| | | MSE | 1.181 | 0.843 | 0.689 | **0.632** | 0.790 | 0.685 | 0.722 | 0.691 | 0.732 | 0.675 |
| | 768 | MAE | 0.670 | 0.460 | 0.401 | 0.404 | – | 0.374 | 0.385 | 0.376 | 0.413 | **0.371** |
| | | MSE | 1.182 | 0.832 | 0.702 | **0.655** | – | 0.688 | 0.732 | 0.687 | 0.740 | 0.680 |

Table 21: Comparison of forecasting performance of S4M (Ours) and baselines on four datasets with various look-back window length under time point missing scenario when missing ratio $r = 0.12$.

| Data | $\ell_L$ | Metric ↓ | BRITS | GRU-D | Trans. | Auto. | BiTGraph | S4 (Mean) | S4 (Ffill) | S4 (Decay) | S4 (SAITS) | S4M (Ours) |
|---|---|---|---|---|---|---|---|---|---|---|---|---|
| Electricity | 96 | MAE | 0.655 | 0.458 | 0.419 | 0.405 | 0.414 | 0.429 | 0.450 | 0.421 | 0.478 | **0.402** |
| | | MSE | 0.666 | 0.394 | 0.339 | **0.313** | 0.331 | 0.369 | 0.390 | 0.347 | 0.442 | 0.328 |
| | 192 | MAE | 0.652 | 0.450 | 0.411 | 0.378 | 0.401 | 0.398 | 0.404 | 0.399 | 0.414 | **0.374** |
| | | MSE | 0.666 | 0.386 | 0.326 | **0.279** | 0.307 | 0.320 | 0.317 | 0.307 | 0.340 | 0.281 |
| | 384 | MAE | 0.659 | 0.450 | 0.430 | 0.395 | 0.437 | 0.395 | 0.398 | 0.397 | 0.413 | **0.378** |
| | | MSE | 0.682 | 0.383 | 0.344 | 0.301 | 0.347 | 0.317 | 0.310 | 0.304 | 0.336 | **0.286** |
| | 768 | MAE | 0.659 | 0.450 | 0.432 | 0.390 | 0.437 | 0.397 | 0.397 | 0.395 | 0.413 | **0.382** |
| | | MSE | 0.680 | 0.384 | 0.348 | 0.299 | 0.347 | 0.319 | 0.310 | 0.303 | 0.337 | **0.299** |
| ETTh1 | 96 | MAE | 0.733 | 0.680 | 0.701 | 0.675 | **0.566** | 0.673 | 0.663 | 0.637 | 0.752 | 0.591 |
| | | MSE | 0.983 | 0.853 | 0.909 | 0.831 | **0.627** | 0.830 | 0.810 | 0.749 | 1.004 | 0.674 |
| | 192 | MAE | 0.759 | 0.687 | 0.686 | 0.662 | 0.628 | 0.616 | 0.615 | 0.605 | 0.711 | **0.595** |
| | | MSE | 1.022 | 0.865 | 0.906 | 0.780 | 0.764 | 0.695 | 0.691 | 0.675 | 0.883 | **0.648** |
| | 384 | MAE | 0.764 | 0.689 | 0.713 | 0.669 | 0.678 | 0.608 | 0.614 | 0.613 | 0.701 | **0.588** |
| | | MSE | 1.042 | 0.869 | 0.949 | 0.794 | 0.905 | 0.679 | 0.686 | 0.688 | 0.841 | **0.638** |
| | 768 | MAE | 0.793 | 0.701 | 1.099 | 0.663 | 0.654 | 0.711 | 0.624 | 0.633 | 0.711 | **0.611** |
| | | MSE | 1.078 | 0.890 | 1.118 | 0.768 | 0.802 | 0.863 | 0.709 | 0.729 | 0.863 | **0.663** |
| Weather | 96 | MAE | 0.413 | 0.402 | 0.471 | 0.698 | 0.556 | 0.401 | 0.385 | 0.385 | 0.536 | **0.355** |
| | | MSE | 0.341 | 0.336 | 0.492 | 0.767 | 0.562 | 0.348 | 0.323 | 0.325 | 0.540 | **0.278** |
| | 192 | MAE | 0.426 | 0.377 | 0.468 | 0.706 | 0.455 | 0.368 | 0.344 | 0.348 | 0.447 | **0.335** |
| | | MSE | 0.365 | 0.303 | 0.414 | 0.789 | 0.431 | 0.299 | 0.271 | 0.275 | 0.386 | **0.253** |
| | 384 | MAE | 0.454 | 0.383 | 0.451 | 0.708 | 0.656 | 0.359 | 0.343 | 0.337 | 0.453 | **0.331** |
| | | MSE | 0.405 | 0.313 | 0.396 | 0.806 | 0.699 | 0.286 | 0.266 | 0.263 | 0.393 | **0.256** |
| | 768 | MAE | 0.480 | 0.382 | 0.418 | 0.719 | 0.633 | 0.364 | 0.340 | 0.336 | 0.446 | 0.350 |
| | | MSE | 0.439 | 0.312 | 0.362 | 0.838 | 0.673 | 0.288 | 0.264 | **0.261** | 0.381 | 0.276 |
| Traffic | 96 | MAE | 0.693 | 0.531 | 0.464 | 0.516 | 0.554 | 0.469 | 0.495 | 0.463 | 0.527 | **0.454** |
| | | MSE | 1.221 | 0.915 | 0.812 | 0.850 | 1.025 | 0.827 | 0.872 | **0.806** | 0.951 | 0.841 |
| | 192 | MAE | 0.685 | 0.521 | 0.448 | 0.413 | 0.501 | 0.401 | 0.441 | 0.419 | 0.426 | **0.397** |
| | | MSE | 1.201 | 0.904 | 0.779 | **0.667** | 0.835 | 0.716 | 0.744 | 0.720 | 0.756 | 0.703 |
| | 384 | MAE | 0.686 | 0.576 | 0.499 | 0.445 | 0.533 | 0.394 | 0.439 | 0.397 | 0.416 | **0.393** |
| | | MSE | 1.222 | 0.962 | 0.839 | 0.744 | 0.908 | **0.692** | 0.740 | 0.694 | 0.731 | 0.702 |
| | 768 | MAE | 0.688 | 0.563 | 0.486 | 0.530 | – | **0.386** | 0.425 | 0.397 | 0.415 | 0.389 |
| | | MSE | 1.226 | 0.949 | 0.801 | 0.920 | – | **0.687** | 0.722 | 0.694 | 0.732 | 0.709 |

Table 22: Comparison of forecasting performance of S4M (Ours) and baselines on four datasets with various look-back window length under variable missing scenario when missing ratio $r = 0.12$.

| Data | $\ell_L$ | Metric ↓ | BRITS | GRU-D | Trans. | Auto. | BiTGraph | S4 (Mean) | S4 (Ffill) | S4 (Decay) | S4 (SAITS) | S4M (Ours) |
|---|---|---|---|---|---|---|---|---|---|---|---|---|
| Electricity | 96 | MAE | 0.641 | 0.452 | 0.402 | 0.395 | 0.426 | 0.396 | 0.403 | 0.410 | 0.503 | 0.387 |
| | | MSE | 0.642 | 0.395 | 0.320 | 0.300 | 0.343 | 0.324 | 0.329 | 0.337 | 0.454 | 0.307 |
| | 192 | MAE | 0.644 | 0.432 | 0.412 | 0.368 | 0.407 | 0.374 | 0.373 | 0.391 | 0.465 | 0.362 |
| | | MSE | 0.649 | 0.368 | 0.331 | 0.262 | 0.315 | 0.284 | 0.281 | 0.304 | 0.388 | 0.270 |
| | 384 | MAE | 0.625 | 0.453 | 0.413 | 0.390 | 0.405 | 0.372 | 0.376 | 0.391 | 0.462 | 0.364 |
| | | MSE | 0.619 | 0.396 | 0.329 | 0.291 | 0.309 | 0.279 | 0.283 | 0.304 | 0.459 | 0.271 |
| | 768 | MAE | 0.643 | 0.466 | 0.442 | 0.369 | 0.397 | 0.377 | 0.381 | 0.385 | 0.381 | 0.365 |
| | | MSE | 0.649 | 0.412 | 0.363 | 0.266 | 0.298 | 0.288 | 0.296 | 0.291 | 0.758 | 0.274 |
| ETTh1 | 96 | MAE | 0.727 | 0.646 | 0.611 | 0.625 | 0.599 | 0.678 | 0.684 | 0.642 | 1.000 | 0.590 |
| | | MSE | 0.960 | 0.778 | 0.687 | 0.718 | 0.707 | 0.836 | 0.830 | 0.766 | 0.718 | 0.651 |
| | 192 | MAE | 0.754 | 0.643 | 0.683 | 0.611 | 0.640 | 0.601 | 0.637 | 0.603 | 0.920 | 0.581 |
| | | MSE | 0.995 | 0.772 | 0.877 | 0.674 | 0.807 | 0.625 | 0.726 | 0.670 | 0.699 | 0.610 |
| | 384 | MAE | 0.757 | 0.645 | 0.626 | 0.662 | 0.623 | 0.623 | 0.607 | 0.605 | 0.868 | 0.594 |
| | | MSE | 1.012 | 0.781 | 0.687 | 0.791 | 0.765 | 0.648 | 0.664 | 0.673 | 0.702 | 0.642 |
| | 768 | MAE | 0.784 | 0.656 | 0.802 | 0.665 | 0.656 | 0.698 | 0.621 | 0.625 | 0.873 | 0.635 |
| | | MSE | 1.045 | 0.792 | 1.061 | 0.787 | 0.810 | 0.848 | 0.701 | 0.726 | 15.503 | 0.721 |
| Weather | 96 | MAE | 0.384 | 0.371 | 0.417 | 0.678 | 0.530 | 0.393 | 0.394 | 0.389 | 0.444 | 0.350 |
| | | MSE | 0.314 | 0.305 | 0.353 | 0.749 | 0.530 | 0.348 | 0.336 | 0.332 | 0.401 | 0.276 |
| | 192 | MAE | 0.397 | 0.362 | 0.425 | 0.684 | 0.433 | 0.362 | 0.350 | 0.347 | 0.421 | 0.322 |
| | | MSE | 0.340 | 0.290 | 0.363 | 0.764 | 0.404 | 0.294 | 0.274 | 0.275 | 0.360 | 0.244 |
| | 384 | MAE | 0.428 | 0.354 | 0.386 | 0.691 | 0.626 | 0.359 | 0.344 | 0.338 | 0.427 | 0.342 |
| | | MSE | 0.379 | 0.282 | 0.316 | 0.789 | 0.663 | 0.291 | 0.268 | 0.265 | 0.365 | 0.264 |
| | 768 | MAE | 0.445 | 0.359 | 0.392 | 0.699 | 0.605 | 0.359 | 0.348 | 0.336 | 0.417 | 0.332 |
| | | MSE | 0.402 | 0.286 | 0.337 | 0.818 | 0.638 | 0.290 | 0.270 | 0.260 | 0.351 | 0.250 |
| Traffic | 96 | MAE | 0.686 | 0.502 | 0.433 | 0.447 | 0.519 | 0.457 | 0.455 | 0.459 | 0.630 | 0.447 |
| | | MSE | 1.232 | 0.955 | 0.750 | 0.727 | 0.924 | 0.834 | 0.875 | 0.882 | 1.082 | 0.867 |
| | 192 | MAE | 0.681 | 0.542 | 0.430 | 0.398 | 0.540 | 0.389 | 0.392 | 0.410 | 0.542 | 0.387 |
| | | MSE | 1.221 | 1.047 | 0.753 | 0.661 | 0.948 | 0.703 | 0.744 | 0.795 | 0.891 | 0.725 |
| | 384 | MAE | 0.683 | 0.534 | 0.415 | 0.406 | 0.485 | 0.387 | 0.392 | 0.405 | 0.558 | 0.387 |
| | | MSE | 1.229 | 1.019 | 0.730 | 0.693 | 0.811 | 0.692 | 0.744 | 0.786 | 0.901 | 0.726 |
| | 768 | MAE | 0.684 | 0.540 | 0.491 | 0.416 | – | 0.387 | 0.389 | 0.398 | 0.541 | 0.400 |
| | | MSE | 1.228 | 1.036 | 0.817 | 0.706 | – | 0.695 | 0.740 | 0.763 | 0.885 | 0.749 |

Table 23: Comparison of forecasting performance of S4M (Ours) and baselines on four datasets with various look-back window length under time point missing scenario when missing ratio $r = 0.24$.

| Data | $\ell_L$ | Metric ↓ | BRITS | GRU-D | Trans. | Auto. | BiTGraph | S4 (Mean) | S4 (Ffill) | S4 (Decay) | S4 (SAITS) | S4M (Ours) |
|---|---|---|---|---|---|---|---|---|---|---|---|---|
| Electricity | 96 | MAE | 0.673 | 0.481 | 0.422 | 0.441 | 0.436 | 0.556 | 0.501 | 0.460 | 0.556 | 0.418 |
| | | MSE | 0.698 | 0.437 | 0.344 | 0.363 | 0.367 | 0.570 | 0.479 | 0.409 | 0.570 | 0.366 |
| | 192 | MAE | 0.681 | 0.456 | 0.434 | 0.453 | 0.410 | 0.464 | 0.410 | 0.420 | 0.464 | 0.391 |
| | | MSE | 0.713 | 0.394 | 0.356 | 0.396 | 0.322 | 0.409 | 0.324 | 0.336 | 0.409 | 0.305 |
| | 384 | MAE | 0.656 | 0.464 | 0.432 | 0.418 | 0.425 | 0.472 | 0.420 | 0.424 | 0.472 | 0.389 |
| | | MSE | 0.671 | 0.404 | 0.357 | 0.343 | 0.340 | 0.417 | 0.334 | 0.341 | 0.417 | 0.304 |
| | 768 | MAE | 0.665 | 0.464 | 0.447 | 0.433 | 0.823 | 0.469 | 0.413 | 0.415 | 0.469 | 0.399 |
| | | MSE | 0.690 | 0.406 | 0.376 | 0.356 | 0.998 | 0.413 | 0.328 | 0.331 | 0.413 | 0.318 |
| ETTh1 | 96 | MAE | 0.765 | 0.733 | 0.695 | 0.749 | 0.654 | 0.710 | 0.717 | 0.681 | 0.841 | 0.627 |
| | | MSE | 1.043 | 0.992 | 0.898 | 0.976 | 0.851 | 0.908 | 0.946 | 0.879 | 1.145 | 0.742 |
| | 192 | MAE | 0.776 | 0.739 | 0.685 | 0.707 | 0.650 | 0.644 | 0.659 | 0.640 | 0.817 | 0.609 |
| | | MSE | 1.047 | 1.004 | 0.893 | 0.856 | 0.815 | 0.739 | 0.792 | 0.782 | 1.076 | 0.703 |
| | 384 | MAE | 0.772 | 0.738 | 0.702 | 0.712 | 0.677 | 0.632 | 0.648 | 0.648 | 0.814 | 0.628 |
| | | MSE | 1.058 | 1.001 | 0.917 | 0.870 | 0.908 | 0.710 | 0.768 | 0.779 | 1.059 | 0.710 |
| | 768 | MAE | 0.800 | 0.744 | 0.793 | 0.702 | 0.630 | 0.639 | 0.661 | 0.672 | 0.801 | 0.632 |
| | | MSE | 1.087 | 1.007 | 1.067 | 0.825 | 0.738 | 0.714 | 0.800 | 0.827 | 1.018 | 0.744 |
| Weather | 96 | MAE | 0.448 | 0.397 | 0.606 | 1.022 | 0.585 | 0.421 | 0.381 | 0.378 | 0.710 | 0.362 |
| | | MSE | 0.389 | 0.328 | 0.602 | 1.571 | 0.598 | 0.379 | 0.321 | 0.317 | 0.866 | 0.286 |
| | 192 | MAE | 0.459 | 0.372 | 0.593 | 1.034 | 0.488 | 0.386 | 0.357 | 0.353 | 0.610 | 0.350 |
| | | MSE | 0.413 | 0.296 | 0.604 | 1.615 | 0.473 | 0.324 | 0.283 | 0.282 | 0.644 | 0.269 |
| | 384 | MAE | 0.489 | 0.375 | 0.563 | 1.024 | 0.656 | 0.381 | 0.349 | 0.343 | 0.607 | 0.358 |
| | | MSE | 0.451 | 0.303 | 0.562 | 1.594 | 0.697 | 0.315 | 0.273 | 0.270 | 0.638 | 0.276 |
| | 768 | MAE | 0.517 | 0.375 | 0.512 | 1.017 | 0.645 | 0.381 | 0.351 | 0.342 | 0.584 | 0.375 |
| | | MSE | 0.489 | 0.304 | 0.490 | 1.586 | 0.683 | 0.312 | 0.276 | 0.268 | 0.592 | 0.300 |
| Traffic | 96 | MAE | 0.705 | 0.641 | 0.490 | 0.607 | 0.554 | 0.487 | 0.569 | 0.529 | 0.658 | 0.485 |
| | | MSE | 1.300 | 1.142 | 0.920 | 1.073 | 1.025 | 0.910 | 1.063 | 0.984 | 1.282 | 0.933 |
| | 192 | MAE | 0.695 | 0.617 | 0.512 | 0.472 | 0.533 | 0.442 | 0.480 | 0.452 | 0.539 | 0.433 |
| | | MSE | 1.267 | 1.110 | 0.950 | 0.804 | 0.949 | 0.826 | 0.870 | 0.812 | 1.014 | 0.787 |
| | 384 | MAE | 0.698 | 0.623 | 0.487 | 0.466 | 0.541 | 0.431 | 0.456 | 0.440 | 0.547 | 0.433 |
| | | MSE | 1.274 | 1.133 | 0.896 | 0.802 | 0.952 | 0.795 | 0.842 | 0.809 | 1.031 | 0.788 |
| | 768 | MAE | 0.700 | 0.628 | 0.509 | 0.463 | – | 0.432 | 0.449 | 0.434 | 0.560 | 0.429 |
| | | MSE | 1.270 | 1.158 | 0.872 | 0.798 | – | 0.799 | 0.823 | 0.789 | 1.030 | 0.789 |

Table 24: Comparison of forecasting performance of S4M (Ours) and baselines on four datasets with various look-back window length under variable missing scenario when missing ratio $r = 0.24$.

| Data | $\ell_L$ | Metric↓ | BRITS | GRU-D | Trans. | Auto. | BiTGraph | S4 (Mean) | S4 (Ffill) | S4 (Decay) | S4 (SAITS) | S4M (Ours) |
|---|---|---|---|---|---|---|---|---|---|---|---|---|
| Electricity | 96 | MAE | 0.647 | 0.497 | 0.424 | 0.423 | 0.436 | 0.407 | 0.431 | 0.430 | 0.621 | **0.402** |
| | | MSE | 0.654 | 0.453 | 0.346 | 0.342 | 0.362 | 0.340 | 0.364 | 0.367 | 0.646 | **0.324** |
| | 192 | MAE | 0.649 | 0.454 | 0.423 | 0.412 | 0.425 | 0.382 | 0.391 | 0.401 | 0.575 | **0.373** |
| | | MSE | 0.659 | 0.388 | 0.348 | 0.326 | 0.341 | 0.299 | 0.301 | 0.316 | 0.557 | **0.281** |
| | 384 | MAE | 0.652 | 0.482 | 0.424 | 0.416 | 0.446 | 0.383 | 0.392 | 0.413 | 0.573 | **0.377** |
| | | MSE | 0.667 | 0.434 | 0.347 | 0.335 | 0.358 | 0.299 | 0.298 | 0.329 | 0.557 | **0.290** |
| | 768 | MAE | 0.654 | 0.509 | 0.469 | 0.407 | 0.415 | **0.380** | 0.398 | 0.410 | 0.569 | 0.383 |
| | | MSE | 0.672 | 0.473 | 0.413 | 0.320 | 0.320 | **0.293** | 0.311 | 0.324 | 0.549 | 0.298 |
| ETTh1 | 96 | MAE | 0.757 | 0.682 | 0.654 | 0.712 | 0.637 | 0.708 | 0.728 | 0.671 | 0.828 | **0.622** |
| | | MSE | 1.016 | 0.874 | **0.742** | 0.875 | 0.807 | 0.916 | 0.959 | 0.847 | 1.161 | 0.766 |
| | 192 | MAE | 0.768 | 0.681 | 0.658 | 0.705 | 0.663 | 0.655 | 0.692 | 0.637 | 0.775 | **0.601** |
| | | MSE | 1.025 | 0.871 | 0.775 | 0.836 | 0.867 | 0.776 | 0.873 | 0.765 | 1.022 | **0.654** |
| | 384 | MAE | 0.774 | 0.681 | 0.630 | 0.691 | 0.661 | 0.648 | 0.657 | 0.669 | 0.753 | **0.630** |
| | | MSE | 1.061 | 0.879 | **0.708** | 0.806 | 0.868 | 0.767 | 0.785 | 0.843 | 0.961 | 0.713 |
| | 768 | MAE | 0.798 | 0.692 | 0.746 | 0.687 | **0.660** | 0.665 | 0.682 | 0.677 | 0.750 | 0.682 |
| | | MSE | 1.072 | 0.895 | 1.004 | **0.782** | 0.868 | 0.808 | 0.842 | 0.852 | 0.955 | 0.829 |
| Weather | 96 | MAE | 0.430 | 0.396 | 0.529 | 0.544 | 0.584 | 0.442 | 0.386 | 0.384 | 0.544 | **0.370** |
| | | MSE | 0.373 | 0.327 | 0.504 | 0.538 | 0.595 | 0.403 | 0.318 | 0.318 | 0.538 | **0.288** |
| | 192 | MAE | 0.454 | 0.385 | 0.514 | 0.505 | 0.490 | 0.385 | 0.355 | 0.356 | 0.505 | **0.355** |
| | | MSE | 0.405 | 0.309 | 0.484 | 0.468 | 0.473 | 0.324 | 0.272 | 0.276 | 0.468 | **0.270** |
| | 384 | MAE | 0.485 | 0.376 | 0.479 | 0.506 | 0.655 | 0.385 | 0.351 | **0.348** | 0.506 | 0.359 |
| | | MSE | 0.443 | 0.300 | 0.436 | 0.469 | 0.693 | 0.320 | **0.269** | 0.269 | 0.469 | 0.278 |
| | 768 | MAE | 0.492 | 0.379 | 0.461 | 0.494 | 0.640 | 0.384 | 0.356 | **0.345** | 0.494 | 0.377 |
| | | MSE | 0.459 | 0.305 | 0.418 | 0.583 | 0.674 | 0.317 | 0.273 | **0.264** | 0.447 | 0.301 |
| Traffic | 96 | MAE | 0.699 | 0.575 | 0.507 | 0.464 | 0.547 | **0.462** | 0.507 | 0.524 | 0.725 | 0.473 |
| | | MSE | 1.266 | 1.074 | 0.891 | **0.778** | 0.998 | 0.850 | 0.977 | 0.969 | 1.245 | 0.896 |
| | 192 | MAE | 0.689 | 0.645 | 0.481 | 0.427 | 0.546 | **0.404** | 0.412 | 0.442 | 0.640 | 0.410 |
| | | MSE | 1.241 | 1.203 | 0.827 | **0.734** | 0.971 | 0.747 | 0.755 | 0.831 | 1.114 | 0.747 |
| | 384 | MAE | 0.690 | 0.643 | 0.509 | 0.428 | 0.483 | **0.401** | 0.408 | 0.435 | 0.641 | 0.414 |
| | | MSE | 1.245 | 1.199 | 0.857 | 0.748 | 0.813 | **0.741** | 0.742 | 0.823 | 1.109 | 0.753 |
| | 768 | MAE | 0.692 | 0.639 | 0.526 | 0.434 | – | **0.389** | 0.408 | 0.436 | 0.633 | 0.438 |
| | | MSE | 1.247 | 1.174 | 0.906 | 0.714 | – | **0.713** | 0.750 | 0.826 | 1.102 | 0.796 |

## D.7 VISUALIZATION

To provide a clear comparison among different models, we list supplementary prediction showcases of Electricity dataset in Fig. 6. The results demonstrate that S4M surpasses other methods in capturing future trends in the presence of missing observations.

## D.8 REAL WORLD APPLICATION

For more general evaluation, we also include the real world dataset, USHCN climate dataset (Menne et al., 2015), with 271728 time steps and 10 variables in total. We set the lookback window $\ell_L = 96$ and horizon window $\ell_H = 96$. The results shown in Tab. 25 further confirm that S4M outperforms other methods on the real-world dataset.

Table 25: Comparison of forecasting performance of S4M (Ours) and baselines on USHCN climate dataset for fixed look-back length and horizon window under time point missing scenario when missing ratio $r = 0.12$.

| $r$ | Metric↓ | BRITS | GRU-D | S4(Mean) | S4(Ffill) | S4(Decay) | S4M(Ours) |
|---|---|---|---|---|---|---|---|
| 0.12 | MAE | 0.644 | 0.477 | 0.477 | 0.489 | 0.466 | **0.447** |
| | MSE | 0.668 | 0.452 | 0.455 | 0.414 | 0.447 | 0.417 |
| 0.24 | MAE | 0.644 | 0.499 | 0.507 | 0.522 | 0.502 | **0.473** |
| | MSE | 0.689 | 0.484 | 0.503 | 0.517 | 0.503 | 0.433 |

| $r$ | Metric↓ | Transformer | Autoformer | BiTGraph | iTransformer | CARD |
|---|---|---|---|---|---|---|
| 0.12 | MAE | 0.461 | 0.511 | 0.474 | 0.478 | 0.451 |
| | MSE | **0.406** | 0.499 | 0.439 | 0.460 | 0.411 |
| 0.24 | MAE | 0.475 | 0.534 | 0.495 | 0.504 | 0.477 |
| | MSE | **0.403** | 0.530 | 0.469 | 0.502 | 0.444 |

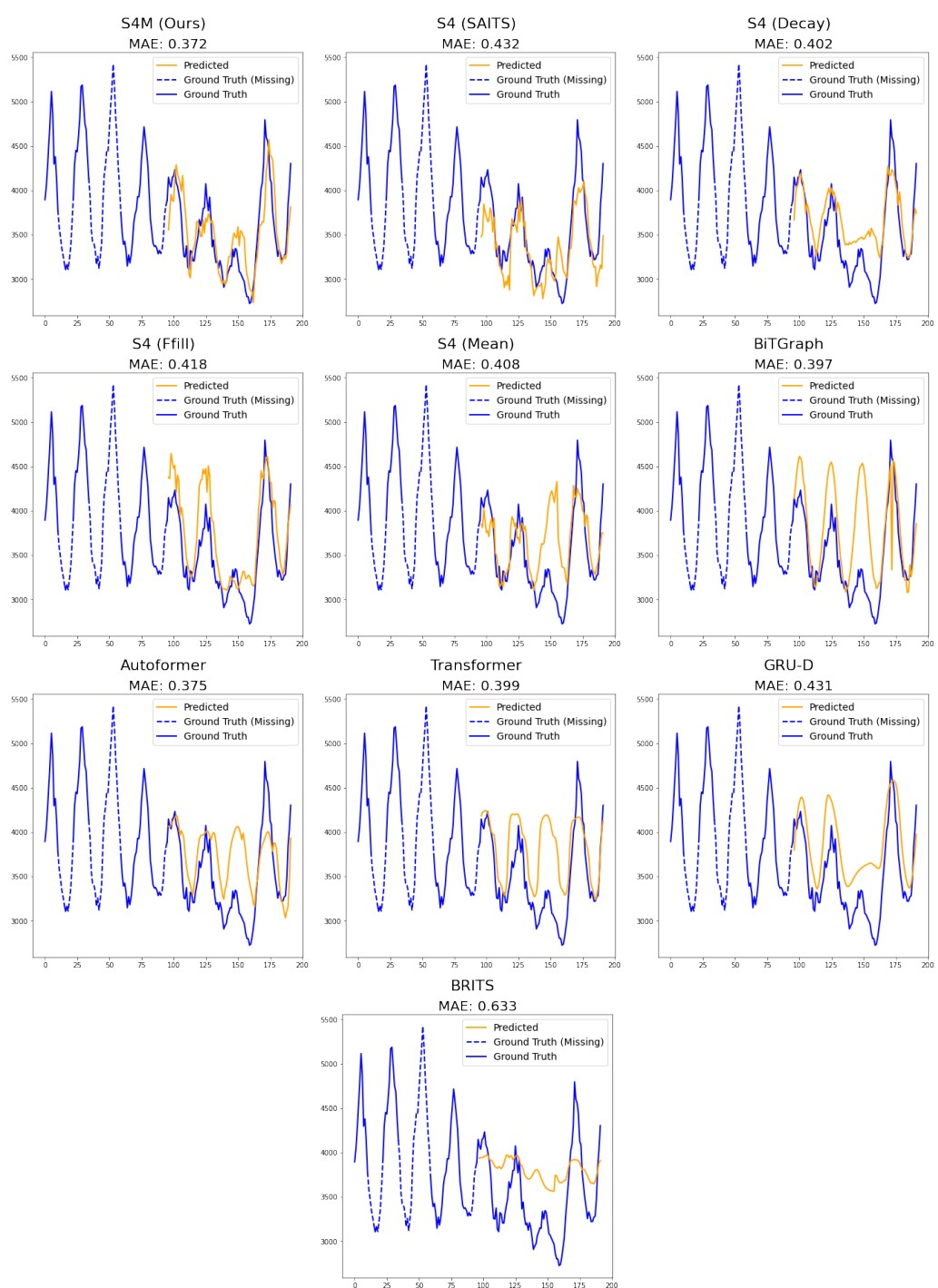

Figure 6: Visualization of S4M (Ours) and baselines with $\ell_H = 96$, $\ell_L = 96$ on the Electricity dataset under time point missing scenario when missing ratio $r = 0.06$.

