# OpenReview forum: "S4M: S4 for multivariate time series forecasting with Missing values"
_ICLR.cc/2025/Conference — ICLR 2025 Poster_

### Official Review · Reviewer_EZj4 · 2024-10-31

**Soundness:** 3
**Presentation:** 3
**Contribution:** 3
**Rating:** 5
**Confidence:** 3

**Summary:**

This paper explicitly models missing patterns within the Structured State Space Sequence (S4) architecture, developing two key modules: the Adaptive Temporal Prototype Mapper and the Missing-Aware Dual Stream. The experiments demonstrate that S4M achieves state-of-the-art performance.

**Strengths:**

1. This paper explores the integration of State Space Models with missing patterns for long-term time series forecasting.

2. The proposed adaptive temporal prototype mapper and missing aware dual stream S4 modules effectively capture rich historical patterns and learn robust representations.

3. Experimental results illustrate that the proposed model achieves state-of-the-art performance in handling missing data.

**Weaknesses:**

1. The authors should further explain the motivation for introducing $\mathbf{\bar{E}}E_m(m_t;\theta_m)$ to the SSM model in Equation 4.

2. This paper lacks a discussion between the S4M and existing methods designed for handling missing values, which diminishes the significance of the proposed model.

3. The settings of hyperparameters $K_1$, $K_2$, $\tau_1$, and $\tau_2$ are emprical. The authors should provide guidance on how to set these hyperparameters across different datasets with varying characteristics.

**Questions:**

1. The proposed model struggles when the input length is shorter than the output length. In addition, it is general to fix the lookback length and adapt to various prediction lengths [1]. Thus, it would strengthen this paper to add experimental results with various horizons.

2. Baselines such as Transformer and Autoformer are designed for complete data. The authors should clarify how these models can be adapted for partially observed data. Besides, the experimental analysis should include state-of-the-art baselines specifically designed for long-term forecasting tasks, such as iTransformer [1], CARD [2], and Crossformer [3].

[1]  Liu Y, Hu T, Zhang H, et al. itransformer: Inverted transformers are effective for time series forecasting[C]. The eleventh international conference on learning representations. 2023.

[2]  Wang X, Zhou T, Wen Q, et al. CARD: Channel aligned robust blend transformer for time series forecasting[C]. The Twelfth International Conference on Learning Representations. 2024.

[3]  Zhang Y, Yan J. Crossformer: Transformer utilizing cross-dimension dependency for multivariate time series forecasting[C]. The eleventh international conference on learning representations. 2023.

---

> ### Author Response · Authors · 2024-11-25
> **Response Part 1**
>
> We thank the reviewer for the thoughtful and detailed review. Also, we appreciate that the reviewer acknowledges our proposed model achieves SOTA performance in handling missing data. We address the raised concerns below.
>
> ***W1. The authors should further explain the motivation for introducing $\overline{\mathbf{E}} E_m(m_{t};\theta_m)$  to the SSM model in Equation 4.***
>
> Thank you for your question about our motivation for incorporating $\overline{\mathbf{E}} E_m(m_{t};\theta_m)$ into the model.
>
> To address the missing data problem in S4 models, we aim to (1) distinguish the missing time points, enabling the model to treat them differently from the observed data (e.g., by referring to data in the prototype bank), and (2) ensure that the core properties of the S4 model are preserved. To this end,  we seek a term that can flag missing values while preserving the HiPPO structure of S4. We found that integrating additional masking terms $M$, inspired by literature [1], to serve as a simple yet effective indicator for the model to recognize missing values. However, since the elements of $M$ take binary values (0 or 1), they are not naturally on the same scale as the other terms in (4). To address this, we designed an encoder to transform the mask information to an appropriate scale. Incorporating this term still preserves the HiPPO structure of S4, thereby enriching the model with additional information while maintaining its core advantages.
>
> ***W2. Highlighting the existing discussion between the S4M and existing methods designed for handling missing values***
>
> In fact, we discussed the differences between our methods v.s. the other existing methods for handling missing values in both Introduction (see lines 62-73) and Appendix A.2. To recap, traditional approaches for handling missing values use a two-step process: imputing missing values first and then performing standard analysis. This can lead to errors and suboptimal results, especially in multivariate time series with complex missing patterns and high missing ratios. We also refer to methods that directly forecast with missing data. RNN-based methods, such as BRITS and GRUD, typically require long training times and exhibit inferior forecasting performance. Graph network-based models, like BiTGraph, are effective at navigating temporal dependencies and spatial structures but often suffer from high memory usage. ODE-based methods, such as Neural ODE, generally incur high computational costs.
>
> In contrast, we propose S4M, which combines a prototype bank with a structured state space model (S4). Our approach focuses on recognizing and representing missing data patterns in the latent space, thereby enhancing model performance by better capturing underlying dependencies while maintaining the high performance of S4.

---

> ### Author Response · Authors · 2024-11-25
> **Response Part 2**
>
> ***W3. Highlighting the existing discussion on  hyperparameters K1, K2, tao1, tao2..***
>
>  We agree on the importance of discussing hyperparameters. Therefore, in our original submission, we provided extensive sensitivity analysis on these hyperparameters in Appendix D.  Specifically, we discussed $K_1$ in, $K_2$ in Appendix D.4.1.  According to our results, a suggested $K_2$ is between 5 and 10, as this range is effective and relatively insensitive to performance. The recommended ranges for $\tau_1$ and $\tau_2$ are [0.3, 0.6] and [0.8, 1.0), respectively. Both parameters can be selected using the validation set. Most of the experiments are robust to the choice of $K_1=30$ or $K_1=50$. If one finds the dataset exists large amout of clusters over 100 (by running a preliminary experiment with a large value of $K_1$ and let the data adaptively tells the number of clusters), they can also adjust this number accordingly.
>
> *Q1. **Adding experimental results with various horizons.***
>
> Thank you for your question. In response to your comment, we have included new results with a fixed lookback window of size 192 and various prediction lengths. The results show that our model **retains top performance** across most horizons for the Traffic, ETTh1, and Weather datasets. Specifically, for the Electricity dataset, our proposed model ranks among the top two performers in most cases.
>
> |  | Horizon Window | Metric | S4(Mean) | S4(FFill) | S4(Decay) | BRITS | GRUD | Transformer | Autoformer | BiTGraph | iTransformer | PatchTST | S4M(Ours) |
> | --- | --- | --- | --- | --- | --- | --- | --- | --- | --- | --- | --- | --- | --- |
> | Electricity | 24 | MAE | 0.392 | 0.406 | 0.396 | 0.624 | 0.442 | 0.412 | 0.457 | 0.361 | 0.415 | 0.349 | 0.369 |
> |  |  | MSE | 0.312 | 0.322 | 0.303 | 0.621 | 0.372 | 0.323 | 0.390 | 0.256 | 0.326 | 0.243 | 0.277 |
> |  | 48 | MAE | 0.389 | 0.400 | 0.397 | 0.658 | 0.393 | 0.424 | 0.437 | 0.370 | 0.421 | 0.357 | 0.377 |
> |  |  | MSE | 0.310 | 0.311 | 0.310 | 0.669 | 0.452 | 0.344 | 0.357 | 0.272 | 0.336 | 0.254 | 0.290 |
> |  | 96 | MAE | 0.464 | 0.410 | 0.420 | 0.681 | 0.456 | 0.434 | 0.453 | 0.410 | 0.427 | 0.365 | 0.381 |
> |  |  | MSE | 0.409 | 0.324 | 0.336 | 0.713 | 0.394 | 0.356 | 0.396 | 0.322 | 0.344 | 0.266 | 0.305 |
> |  | 192 | MAE | 0.424 | 0.462 | 0.465 | 0.776 | 0.508 | 0.451 | 0.443 | 0.444 | 0.430 | 0.375 | 0.405 |
> |  |  | MSE | 0.363 | 0.414 | 0.411 | 1.047 | 0.472 | 0.378 | 0.366 | 0.365 | 0.348 | 0.278 | 0.331 |
> | ETTh1 | 24 | MAE | 0.529 | 0.538 | 0.585 | 0.750 | 0.600 | 0.617 | 0.656 | 0.558 | 0.586 | 0.575 | 0.554 |
> |  |  | MSE | 0.532 | 0.574 | 0.681 | 0.979 | 0.708 | 0.687 | 0.740 | 0.591 | 0.621 | 0.617 | 0.585 |
> |  | 48 | MAE | 0.577 | 0.553 | 0.605 | 0.756 | 0.646 | 0.623 | 0.699 | 0.639 | 0.630 | 0.574 | 0.573 |
> |  |  | MSE | 0.630 | 0.588 | 0.701 | 1.040 | 0.807 | 0.680 | 0.844 | 0.831 | 0.591 | 0.623 | 0.633 |
> |  | 96 | MAE | 0.644 | 0.659 | 0.640 | 0.776 | 0.739 | 0.685 | 0.707 | 0.650 | 0.689 | 0.578 | 0.604 |
> |  |  | MSE | 0.739 | 0.792 | 0.782 | 1.047 | 1.004 | 0.893 | 0.856 | 0.815 | 0.781 | 0.622 | 0.672 |
> |  | 192 | MSE | 0.691 | 0.672 | 0.692 | 0.770 | 0.750 | 0.743 | 0.715 | 0.691 | 0.661 | 0.598 | 0.655 |
> |  |  | MAE | 0.864 | 0.833 | 0.894 | 1.040 | 1.011 | 0.975 | 0.874 | 0.935 | 0.754 | 0.662 | 0.801 |
> | Weather | 24 | MAE | 0.360 | 0.312 | 0.310 | 0.494 | 0.394 | 0.449 | 1.020 | 0.308 | 0.511 | 0.314 | 0.304 |
> |  |  | MSE | 0.288 | 0.224 | 0.222 | 0.438 | 0.314 | 0.376 | 1.565 | 0.234 | 0.468 | 0.231 | 0.212 |
> |  | 48 | MSE | 0.400 | 0.347 | 0.352 | 0.490 | 0.431 | 0.535 | 1.032 | 0.356 | 0.511 | 0.347 | 0.339 |
> |  |  | MAE | 0.340 | 0.264 | 0.271 | 0.440 | 0.358 | 0.485 | 1.604 | 0.277 | 0.467 | 0.268 | 0.250 |
> |  | 96 | MAE | 0.386 | 0.357 | 0.353 | 0.459 | 0.372 | 0.593 | 1.034 | 0.488 | 0.515 | 0.377 | 0.357 |
> |  |  | MSE | 0.324 | 0.283 | 0.282 | 0.413 | 0.296 | 0.604 | 1.615 | 0.473 | 0.470 | 0.303 | 0.276 |
> |  | 192 | MAE | 0.532 | 0.503 | 0.505 | 0.519 | 0.559 | 0.588 | 1.035 | 0.628 | 0.521 | 0.416 | 0.410 |
> |  |  | MSE | 0.538 | 0.485 | 0.489 | 0.489 | 0.561 | 0.586 | 1.626 | 0.664 | 0.479 | 0.351 | 0.386 |
> | Traffic | 24 | MAE | 0.441 | 0.459 | 0.435 | 0.672 | 0.569 | 0.472 | 0.561 | 0.496 | 0.463 | 0.461 | 0.420 |
> |  |  | MSE | 0.787 | 0.833 | 0.788 | 1.207 | 1.082 | 0.821 | 0.966 | 0.496 | 0.696 | 0.713 | 0.762 |
> |  | 48 | MAE | 0.442 | 0.472 | 0.449 | 0.682 | 0.600 | 0.485 | 0.519 | 0.527 | 0.471 | 0.472 | 0.420 |
> |  |  | MSE | 0.825 | 0.831 | 0.806 | 1.220 | 1.104 | 0.871 | 0.889 | 0.930 | 0.718 | 0.739 | 0.709 |
> |  | 96 | MAE | 0.442 | 0.480 | 0.452 | 0.695 | 0.617 | 0.512 | 0.472 | 0.533 | 0.480 | 0.478 | 0.434 |
> |  |  | MSE | 0.826 | 0.870 | 0.812 | 1.267 | 1.110 | 0.950 | 0.804 | 0.949 | 0.746 | 0.763 | 0.810 |
> |  | 192 | MAE | 0.486 | 0.547 | 0.498 | 0.695 | 0.566 | 0.478 | 0.512 | 0.550 | 0.488 | 0.483 | 0.478 |
> |  |  | MSE | 0.869 | 0.992 | 0.901 | 0.695 | 1.037 | 0.861 | 0.866 | 0.997 | 0.766 | 0.770 | 0.886 |

---

> ### Author Response · Authors · 2024-11-25
> **Response Part 3**
>
> ***Q2.1 How did the baseline models  adapted for partially observed data.***
>
> Thank you for your question. In our experiments, the missing values are imputed using the mean value, after which the Transformer and Autoformer models were applied. The same procedure is used for the additional experiments with iTransformer, CARD, and PatchTST.
>
> ***Q2.2  Justification on deliberately omitted methods not designed for non-missing data. Besides, the experimental analysis should include SOTA baselines specifically designed for long-term forecasting tasks, such as iTransformer [1], CARD [2], and Crossformer [3].***
>
> Thank you for raising this issue. In our initial submission, we did not include iTransformer and CARD because these methods are not specifically designed for time series with missing values. Comparing them directly with our method might lead to an unfair evaluation. Instead, we focused on comparisons with SOTA methods tailored for time series prediction with missing values, such as BiTGraph, as well as S4-based methods. For transformer-based baselines, we selected two classic architectures: Transformer and Autoformer.
>
> To provide a more comprehensive evaluationaddress your request, we have now included PatchTST, iTransformer, and CARD in our experiments. The results on four benchmark datasets are presented in the table below, with [additional results](https://openreview.net/forum?id=BkftcwIVmR&noteId=qh9vNw3wyX) on a real-world dataset. Furthermore, we analyze the [computational cost](https://openreview.net/forum?id=BkftcwIVmR&noteId=Elj4yMG1xv) and [the performance](https://openreview.net/forum?id=BkftcwIVmR&noteId=bcEimHMmTq) across different horizon windows.
>
> Among the three additional methods, PatchTST exhibits strong performance in handling missing values, particularly on the Electricity dataset, and also performs well in scenarios without missing values. However, in other settings, its results are less competitive compared to S4M. Additionally, as shown in Table X, PatchTST incurs significantly higher training and inference times than S4M.
>
> | Dataset | Metric | S4M | PatchTST | iTransformer | CARD |
> | --- | --- | --- | --- | --- | --- |
> | Electricity | MAE | 0.418 | 0.420 | 0.452 | 0.440 |
> |  | MSE | 0.359 | 0.344 | 0.389 | 0.366 |
> | ETTh1 | MSE | 0.627 | 0.583 | 0.668 | 0.780 |
> |  | MAE | 0.742 | 0.650 | 0.786 | 1.041 |
> | Weather | MSE | 0.370 | 0.399 | 0.510 | 0.422 |
> |  | MAE | 0.294 | 0.327 | 0.459 | 0.376 |
> | Traffic | MSE | 0.499 | 0.530 | 0.519 | 0.554 |
> |  | MAE | 0.943 | 0.927 | 0.897 | 0.965 |

---

> ### Author Response · Authors · 2024-11-28
>
> Dear reviewer EZj4,
>
> We sincerely thank you for your thoughtful review and valuable feedback. We have carefully addressed each of your questions and provided detailed responses in the rebuttal. We hope to have resolved all your concerns. If you have any further comments, we would be glad to address them before the rebuttal period ends. If our responses address your concerns, we would deeply appreciate it if you could consider raising your score. Your recognition for our novel work means a lot. Thanks again for your time and effort in reviewing our work.
>
> Regards,
> S4M authors

---

> ### Comment · Area_Chair_Dm2o · 2024-12-02
> **Please respond to the authors of submission 10665**
>
> Dear Reviewer EZj4,
>
> We are at the end of the discussion period, so please take some time to read the response to your review for submission 10665
>
> Also, please indicate the extent to which the response addresses your concerns and whether it changes your score - try to explain your decision.
>
> All the best,
>
> The AC

---

### Official Review · Reviewer_iyXX · 2024-11-01

**Soundness:** 2
**Presentation:** 4
**Contribution:** 3
**Rating:** 6
**Confidence:** 4

**Summary:**

In this paper, the authors propose S4M. S4M is an adaption of S4 which can handle missing values by:
- Using prototype clusters, look-back information and an encoder to find representations also for time-points where values are missing
- By explicitly also incorporating the masking matrix M into the S4-Layers.

 They evaluate S4M on the standard data for regular time-series forecasting when some data is hidden and show
 that i) S4M is competitive or often outperforms other S4 and Missing-Value approaches.

**Strengths:**

+ The idea of the prototype bank is very compelling and thoughtful, I like it a lot.
+ The presentation is very good. The paper is written in a manner which makes it comfortable to follow.
  Especially having an algorithm for each of the crucial parts helped me a lot.
+ The results are not looking like fundamental break-throughs, but they are very promising for such a novel approach and there are a lot of ablations studies/hyperparameter experiments.

**Weaknesses:**

The two main weaknesses I identified:

- My largest critique point is, that the authors are not comparing at all with recent results from the "Irregular Sampled Time-Series wit Missing Values" Literature. There is a plentitude of recent works solving irregular time-series forecasting in an end-to-end manner via ODEs, modelling latent dynamics or graph modelling.[1-5]. Furthermore, these papers provide a set of standard datasets for time-series forecasting with missing values, thus no need to synthetically make the normal regular datasets irregular.

- There are a lot of standard methods missing, where one could do simply linear interpolation to use them in the experiments on which S4 is tested. For example, this work is not referring to important forecasting works like PatchTST or iTransformer at all. The results of S4M in Table 1, are way worse then the results of PatchTST (see Table at https://github.com/yuqinie98/PatchTST?tab=readme-ov-file) that it may be the case that PatchTST with 0.06 missing values indeed outperforms S4M.

[1] De Brouwer, E., Simm, J., Arany, A., & Moreau, Y. (2019). GRU-ODE-Bayes: Continuous modeling of sporadically-observed time series. Advances in neural information processing systems, 32.

[2] Yalavarthi, Vijaya Krishna, et al. "GraFITi: Graphs for Forecasting Irregularly Sampled Time Series." Proceedings of the AAAI Conference on Artificial Intelligence. Vol. 38. No. 15. 2024.

[3] Schirmer, Mona, et al. "Modeling irregular time series with continuous recurrent units." International conference on machine learning. PMLR, 2022.

[4] Biloš, Marin, et al. "Neural flows: Efficient alternative to neural ODEs." Advances in neural information processing systems 34 (2021): 21325-21337.

[5] Klötergens, Christian, et al. "Functional Latent Dynamics for Irregularly Sampled Time Series Forecasting." Joint European Conference on Machine Learning and Knowledge Discovery in Databases.

## My Current Rating
I do really like the idea and think that it has potential, even beyond S4 models for irregular time-series forecasting. However, for a top conference like ICLR, the amount of missing comparison to important related work is too high for recommending acceptance.

**Questions:**

Additionally to the critique points mentioned above, I have the following comments/question:

- Have you tested S4M when there are no missing values at all? I would be curious whether your prototype bank is also useful if no values are missing.
- Table 3: Do I understand correctly, that you are comparing: Prototyping Bank + Masking (i.e. having m_t in (4)) against only having the prototype bank? I would also like to see Only Masking, i.e. having the mask m_t in (4) but no prototype bank, i.e. replacing o_t with X_t. Is your prototype-bank really needed for irregular time-series forecasting?

---

> ### Author Response · Authors · 2024-11-25
> **Response Part 1**
>
> We thank the reviewer for the thoughtful and detailed review. Also, we appreciate that the reviewer acknowledges our prototype bank presents an interesting practical way to tackle missing values. We address the raised concerns below.
>
>  ***W1.1 Justification on the comparison baseline selections.***
>
> We thank the reviewer for the question. The reasons that we did not directly compare S4M with irregularly sampled time-series methods are multifold:  First, here, we focus on **block-based missing** patterns where the observed values occur at consecutive time points (see our Fig. 4). The irregularly sampled time series problems typically don’t directly consider the properties of such missing patterns. Second, **efficiency is an essential consideration** for our work, as stated in **lines 51-52** in our original submission.  Compared with S4, the suggested ODE and graph-based irregularly sampled methods are computationally costly.  For instance, experiments with Grafiti on the Traffic and Electricity datasets often result in out-of-memory (OOM) errors in most settings. Similarly, CRU is extremely slow due to its iterative computations over variable dimensions. Detailed comparisons and computational costs for these methods are provided in the table below
>
> | Method | S4(Mean) | S4(Ffill) | S4(Decay) | BRITS | GRUD | Transformer | Autoformer | BiTGraph | iTransformer | PatchTST | CRUD | Grafiti | S4M(Our) |
> | --- | --- | --- | --- | --- | --- | --- | --- | --- | --- | --- | --- | --- | --- |
> | Flops(M) | 12463.39 | 12463.39 | 12618.52 | 9091.16 | 3813.82 | 17627.87 | 18734.88 | 3185.64 | 565.36 | 392299.02 | 219.57 | 265118.32 | 139191.88 |
> | Training Time(s) | 0.11282 | 0.11498 | 0.08325 | 0.46920 | 0.19958 | 0.09035 | 0.09613 | 0.24546 | 0.06744 | 0.46017 | 49.40020 | OOM | 0.219381 |
> | Inference Time(s) | 0.07416 | 0.07983 | 0.06152 | 0.21126 | 0.08756 | 0.06088 | 0.07662 | 0.08122 | 0.04009 | 0.16896 | 4.76765 | OOM | 0.099314 |
>
> In response to your comment, we have two additional SOTA methods for irregular time-series forecasting, Grafiti and CRUD, for comparison. However, due to the high computational costs and significant memory requirements of methods designed for irregularly sampled time-series data, our experiments were restricted to the smaller-scale ETTh1 dataset, as detailed below. The results show that **ODE-based models perform suboptimally** in this context.
>
> | Lookback Window ($L$) | Metric | S4M(Ours) | CRU | Grafiti |
> | --- | --- | --- | --- | --- |
> | 96 | MAE | 0.630 | 0.774 | 0.821 |
> |  | MSE | 0.779 | 1.093 | 1.162 |
> | 192 | MAE | 0.604 | 0.802 | 0.811 |
> |  | MSE | 0.670 | 1.150 | 1.161 |
> | 384 | MAE | 0.6281 | 0.802 | 0.805 |
> |  | MSE | 0.748 | 1.184 | 1.162 |
> | 768 | MAE | 0.619 | 0.774 | 0.821 |
> |  | MSE | 0.693 | 1.093 | 1.060 |

---

> ### Author Response · Authors · 2024-11-25
> **Response Part 2**
>
> ***W1.2 Justification on Datasets Selection.***
>
> There appears to be a misunderstanding on the type of data missing problem we are focusing on in this study. Our work considers cases with block missing patterns in regularly sampled time series, where the observed values occur at consecutive time points (see our Fig. 4). This structure enables the design of an informative representation $o_t$, which is crucial for capturing temporal dependencies effectively. In contrast, standard irregularly sampled time series, like MIMIC and Physionet, do not contain such patterns of consecutive observations in the non-missing time points, making them outside the scope of our study.
>
> Although our current design does not consider the general irregular sampled data, we appreciate your encouraging recognition that "I do really like the idea and think that it has potential, even beyond S4 models for irregular time-series forecasting." We also believe in the benefits of introducing a prototype bank beyond the S4 model, and we hope our work lays the foundation to inspire future work along this valuable future direction.
>
> ***W2.1 Comparison with the linear interpolation method.***
>
> Thank you for your advice. We have included simple and standard imputation methods such as mean, forward fill (Ffill), and linear decay interpolation in their original forms. To incorporate your suggestion, we have added linear interpolation methods to the following table, which presents experiments conducted on four datasets with $r = 0.24$ and a horizon window of 96. The results indicate that, linear interpolation’s performance is **inferior** to both our proposed approach and  the decay method in most cases.
>
> | Dataset | Horizon Length | Metric | S4(Mean) | S4(Ffill) | S4(Decay) | S4(Linear) | S4M(Ours) |
> | --- | --- | --- | --- | --- | --- | --- | --- |
> | Electricity | 96 | MAE | 0.556 | 0.501 | 0.460 | 0.468 | 0.418 |
> |  |  | MSE | 0.570 | 0.479 | 0.409 | 0.425 | 0.366 |
> |  | 192 | MAE | 0.464 | 0.410 | 0.420 | 0.395 | 0.391 |
> |  |  | MSE | 0.409 | 0.324 | 0.336 | 0.306 | 0.305 |
> |  | 384 | MAE | 0.472 | 0.420 | 0.424 | 0.403 | 0.389 |
> |  |  | MSE | 0.417 | 0.334 | 0.341 | 0.311 | 0.304 |
> |  | 768 | MAE | 0.469 | 0.413 | 0.415 | 0.402 | 0.399 |
> |  |  | MSE | 0.413 | 0.328 | 0.331 | 0.315 | 0.318 |
> | ETTh1 | 96 | MAE | 0.710 | 0.717 | 0.681 | 0.696 | 0.627 |
> |  |  | MSE | 0.908 | 0.946 | 0.879 | 0.943 | 0.742 |
> |  | 192 | MAE | 0.644 | 0.659 | 0.640 | 0.671 | 0.609 |
> |  |  | MSE | 0.739 | 0.792 | 0.782 | 0.872 | 0.703 |
> |  | 384 | MAE | 0.632 | 0.648 | 0.648 | 0.646 | 0.628 |
> |  |  | MSE | 0.710 | 0.768 | 0.779 | 0.782 | 0.710 |
> |  | 768 | MAE | 0.639 | 0.661 | 0.672 | 0.659 | 0.632 |
> |  |  | MSE | 0.714 | 0.800 | 0.827 | 0.823 | 0.744 |
> | Weather | 96 | MAE | 0.421 | 0.381 | 0.378 | 0.399 | 0.362 |
> |  |  | MSE | 0.379 | 0.321 | 0.317 | 0.339 | 0.286 |
> |  | 192 | MAE | 0.386 | 0.357 | 0.353 | 0.354 | 0.350 |
> |  |  | MSE | 0.324 | 0.283 | 0.282 | 0.276 | 0.269 |
> |  | 384 | MAE | 0.381 | 0.349 | 0.343 | 0.349 | 0.358 |
> |  |  | MSE | 0.315 | 0.273 | 0.270 | 0.272 | 0.276 |
> |  | 768 | MAE | 0.381 | 0.351 | 0.342 | 0.399 | 0.375 |
> |  |  | MSE | 0.312 | 0.276 | 0.268 | 0.339 | 0.300 |
> | Traffic | 96 | MAE | 0.487 | 0.569 | 0.529 | 0.568 | 0.485 |
> |  |  | MSE | 0.910 | 1.063 | 0.984 | 1.043 | 0.933 |
> |  | 192 | MAE | 0.442 | 0.480 | 0.452 | 0.466 | 0.433 |
> |  |  | MSE | 0.826 | 0.870 | 0.812 | 0.842 | 0.787 |
> |  | 384 | MAE | 0.431 | 0.456 | 0.440 | 0.524 | 0.433 |
> |  |  | MSE | 0.795 | 0.842 | 0.809 | 0.953 | 0.788 |
> |  | 768 | MAE | 0.432 | 0.449 | 0.434 | 0.439 | 0.429 |
> |  |  | MSE | 0.799 | 0.823 | 0.789 | 0.790 | 0.789 |

---

> ### Author Response · Authors · 2024-11-25
> **Response Part 3**
>
> ***W2.2. Justification on MethodsComparison.***
>
> Thank you for your comments on the comparison between PatchTST and our method. There might be a misunderstanding of the comparison. PatchTST does not include straightforward forecasting with missing values. Masking is only used in the context of self-supervised learning. Therefore, the results with 0.06 missing values (~40% missing ratio) are not directly comparable. In our initial submission, we did not include iTransformer and PatchTST because these methods are not specifically designed for time series with missing values. Comparing them directly with our method might lead to an unfair evaluation. Instead, we focused on comparisons with SOTA methods tailored for time series prediction with missing values, such as BiTGraph, as well as S4-based methods. For transformer-based baselines, we selected two representative architectures: Transformer and Autoformer.
>
> To address your request , we have now included PatchTST, iTransformer, and CARD in our experiments. The results on four benchmark datasets are presented in the table below, with additional results on a real-world dataset(https://openreview.net/forum?id=BkftcwIVmR&noteId=qh9vNw3wyX). Furthermore, we analyze the computational cost (https://openreview.net/forum?id=BkftcwIVmR&noteId=8TefCTqULs) and the performance across different horizon windows (https://openreview.net/forum?id=BkftcwIVmR&noteId=bcEimHMmTq).
>
> Among the three additional methods, PatchTST exhibits strong performance in handling missing values, particularly on the Electricity dataset, and also performs well in scenarios without missing values. However, in most of the settings,  S4M achieves consistently superior performance. Additionally, as the table on computational cost(https://openreview.net/forum?id=BkftcwIVmR&noteId=8TefCTqULs) shows, S4M is significantly more efficient than these three suggested methods..
>
> ***Q1. Have you tested S4M when there are no missing values at all? I would be curious whether your prototype bank is also useful if no values are missing.***
>
> Thank you for the question. By design, our method is suitable for time series with block-based missing data. The historical features stored in the prototype bank are particularly helpful when the missing ratio is high. If there is no missing data i, as expected, our method will not show a significant advantage over the other methods but will still maintain a very competitive performance. For this experiment, we report the results using a horizon window of 96 and a lookback window of 96, with no missing values in the original dataset.
>
> | Dataset | Metric | BRITS | GRUD | Transformer | Autoformer | S4 | BiTGraph | S4M(Ours) |
> | --- | --- | --- | --- | --- | --- | --- | --- | --- |
> | Electricity | MAE | 0.398 | 0.413 | 0.411 | 0.352 | 0.386 | 0.348 | 0.383 |
> |  | MSE | 0.318 | 0.332 | 0.321 | 0.242 | 0.301 | 0.254 | 0.295 |
> | ETTh1 | MAE | 0.676 | 0.571 | 0.604 | 0.556 | 0.538 | 0.530 | 0.538 |
> |  | MSE | 0.867 | 0.636 | 0.677 | 0.588 | 0.560 | 0.571 | 0.560 |
> | Weather | MAE | 0.373 | 0.370 | 0.383 | 0.306 | 0.363 | 0.504 | 0.332 |
> |  | MSE | 0.29136 | 0.301 | 0.298 | 0.235 | 0.301 | 0.494 | 0.259 |
> | Traffic | MAE | 0.428 | 0.446 | 0.405 | 0.454 | 0.425 | 0.504 | 0.414 |
> |  | MSE | 0.770 | 0.840 | 0.707 | 0.705 | 0.425 | 0.879 | 0.761 |

---

> ### Author Response · Authors · 2024-11-25
> **Response Part 4**
>
> ***Q2. I would also like to see Only Masking, i.e. having the mask m_t in (4) but no prototype bank, i.e. replacing o_t with X_t. Is your prototype-bank really needed for irregular time-series forecasting?***
>
> Thank you for your valuable feedback. We have included an ablation study for the first module, ATPM. In this study, we compare S4M with and without ATPM, highlighting the improvements brought by ATPM, particularly on the Traffic and ETTh1 datasets. The following experiments were conducted under the same settings as those in the ablation studies presented in the paper.
>
> | Dataset |  | Electricity |  | ETTh1 |  | Weather |  | Traffic |  |
> | --- | --- | --- | --- | --- | --- | --- | --- | --- | --- |
> | $\ell_L$ | Metrics | S4M (Ours) | S4M (w/o prototype) | S4M (Ours) | S4M (w/o prototype) | S4M (Ours) | S4M (w/o prototype) | S4M (Ours) | S4M (w/o prototype) |
> |  |  |  |  | Variable Missing |  |  |  |  |  |
> | 96 | MAE | 0.369 | +0.011 | 0.571 | +0.044 | 0.336 | +0.020 | 0.442 | +0.024 |
> |  | MSE | 0.282 | +0.010 | 0.624 | +0.091 | 0.267 | +0.206 | 0.786 | +0.125 |
> | 192 | MAE | 0.357 | +0.010 | 0.568 | +0.045 | 0.320 | +0.600 | 0.381 | +0.030 |
> |  | MSE | 0.261 | +0.009 | 0.598 | +0.090 | 0.261 | +0.002 | 0.685 | +0.092 |
> | 384 | MAE | 0.359 | +0.009 | 0.584 | +0.029 | 0.334 | +0.006 | 0.383 | +0.026 |
> |  | MSE | 0.264 | +0.009 | 0.613 | +0.064 | 0.256 | +0.008 | 0.700 | +0.065 |
> | 768 | MAE | 0.362 | +0.020 | 0.599 | +0.028 | 0.341 | +0.016 | 0.383 | +0.026 |
> |  | MSE | 0.269 | +0.002 | 0.649 | +0.058 | 0.266 | +0.011 | 0.697 | +0.074 |
> |  |  |  |  | Timepoint Missing |  |  |  |  |  |
> | 96 | MAE | 0.372 | +0.025 | 0.571 | +0.049 | 0.313 | +0.021 | 0.428 | +0.045 |
> |  | MSE | 0.287 | +0.030 | 0.624 | +0.110 | 0.237 | +0.017 | 0.809 | +0.116 |
> | 192 | MAE | 0.367 | +0.004 | 0.574 | +0.039 | 0.305 | +0.006 | 0.385 | +0.005 |
> |  | MSE | 0.274 | +0.004 | 0.593 | +0.110 | 0.225 | +0.001 | 0.687 | +0.023 |
> | 384 | MAE | 0.370 | +0.014 | 0.571 | +0.057 | 0.306 | +0.012 | 0.385 | +0.013 |
> |  | MSE | 0.277 | +0.004 | 0.624 | +0.112 | 0.220 | +0.015 | 0.702 | +0.047 |
> | 768 | MAE | 0.373 | +0.013 | 0.588 | +0.048 | 0.316 | +0.005 | 0.388 | +0.000 |
> |  | MSE | 0.282 | +0.016 | 0.647 | +0.079 | 0.232 | +0.004 | 0.699 | +0.024 |
>
> # *R*

---

> > ### Comment · Reviewer_iyXX · 2024-11-26
> > **Answer to Rebuttal**
> >
> > Dear Authors,
> > thank you for the rebuttal. My concerns are only partially adressed:
> > - I think that the differentiation to irregular sampled time-series has to be made more explicit in the paper. Furthermore, the fact that you are only considering irregular-sampled time-series with specific patterns of missingness is not clear in the current version of the paper.
> > - Your response part 2: My request was more about doing linear interpolation etc and then having models like PatchTST and iTransformer on top, not S4. Because having a look at PatchTST results without missing values, it stands to reason that it outperforms S4M.

---

> ### Author Response · Authors · 2024-11-27
>
> Dear reviwer, thank you for acknowledging that our responses have addressed your concerns in part. We are delighted to engage further and address your remaining questions.
>
> **Clarifications on Time-Series Setting**: In our revision, we have made explicit the distinction between irregularly sampled time-series and our focus on the missing data setting. This distinction is now emphasized both in the abstract and the instructions, ensuring clarity for all readers. Additionally, to guide understanding, we have highlighted Figure 4, which comprehensively illustrates this setting. Thanks for your suggestions. We hope this addresses your concern.
>
> **Focus on S4-based Models**:
> We respectfully request the reviewer to consider that the primary focus of our study is on S4-based models, as stated in our introduction. S4 was selected due to its efficiency, which has been well-documented in the literature. We further demonstrated this efficiency in response to your W2.2 (Part 3), where we compared S4M against the suggested methods and many others. For your convenience, we have included the table from our response, which underscores that **S4M is significantly more efficient than the three suggested methods**.
>
> | Method | S4(Mean) | S4(Ffill) | S4(Decay) | BRITS | GRUD | Transformer | Autoformer | BiTGraph | iTransformer | PatchTST | CRUD | Grafiti | S4M(Our) |
> | --- | --- | --- | --- | --- | --- | --- | --- | --- | --- | --- | --- | --- | --- |
> | Flops(M) | 12463.39 | 12463.39 | 12618.52 | 9091.16 | 3813.82 | 17627.87 | 18734.88 | 3185.64 | 565.36 | 392299.02 | 219.57 | 265118.32 | 139191.88 |
> | Training Time(s) | 0.11282 | 0.11498 | 0.08325 | 0.46920 | 0.19958 | 0.09035 | 0.09613 | 0.24546 | 0.06744 | 0.46017 | 49.40020 | OOM | 0.219381 |
> | Inference Time(s) | 0.07416 | 0.07983 | 0.06152 | 0.21126 | 0.08756 | 0.06088 | 0.07662 | 0.08122 | 0.04009 | 0.16896 | 4.76765 | OOM | 0.099314 |
>
> We hope the reviewer agrees with the trade-offs inherent in various backbone architectures (no free lunch) and recognizes improving S4 is central to this study, reflected in the title and justified in our introduction.
>
>
> **Response to Interpretation of Reviewer Comments**: We appreciate your clarification regarding your request. Initially, we interpreted your comments as two distinct questions, leading us to provide detailed responses in W2.1 (Part 2) and W2.2 (Part 3) separately.
>
> **Experimental Comparison with Your Suggested Methods**: We appreciate your thoughtful feedback. We conducted experiments incorporating your suggested methods with the mean interpolation approach (specified in W2.2, Part 3). Furthermore, we conducted experiments combining PatchTST with linear interpolation. The results are shown below.
> In most settings, S4M consistently demonstrated superior performance. Both linear and mean interpolation with PatchTST do not work well. Linear interpolation outperformed mean interpolation only on the ETTh1 dataset. For datasets exhibiting clear seasonality like electricity and traffic, linear interpolation may perform worse than mean interpolation.
>
> | Dataset | Metric | S4M (Ours) | PatchTST (Linear Interpolation) | PatchTST (Mean Interpolation) |
> | --- | --- | --- | --- | --- |
> | Electricity | MAE | 0.418 | 0.501 | 0.420 |
> |  | MSE | 0.359 | 0.466 | 0.344 |
> | ETTh1 | MAE | 0.627 | 0.587 | 0.583 |
> |  | MSE | 0.742 | 0.649 | 0.650 |
> | Weather | MAE | 0.37 | 0.348 | 0.399 |
> |  | MSE | 0.294 | 0.309 | 0.327 |
> | Traffic | MAE | 0.499 | 0.655 | 0.530 |
> |  | MSE | 0.943 | 0.929 | 0.927 |

---

> > ### Comment · Reviewer_iyXX · 2024-11-28
> > **Reaction To Latest Comment**
> >
> > Dear Authors,
> > thanks for the additional results and changes. I think the modifications and additional experiments strengthen the paper. The authors spend a lot of effort to incorporate the changes I proposed. i thus increased my score.

---

> > > ### Author Response · Authors · 2024-11-28
> > >
> > > Dear reviewer, we really appreciate the time and effort that you have dedicated to providing your valuable feedback on improving our manuscript. We are grateful for your insightful comments. Thank you.

---

### Official Review · Reviewer_iDLq · 2024-11-03

**Soundness:** 2
**Presentation:** 3
**Contribution:** 2
**Rating:** 6
**Confidence:** 3

**Summary:**

The paper introduces S4M, an extension of the S4 framework to multivariate time series forecasting with missing values. It combines a prototype-based representation learning module (ATPM) with a dual-stream S4 architecture (MDS-S4) to handle missing values directly rather than through preprocessing. The method is evaluated on four datasets under various missing data scenarios.

**Strengths:**

The paper addresses a practical and relevant problem. Real-world data often contains missing values (or mis-recorded values) which makes developing principled methods to handle them in forecasting models a worthwhile endeavor. The paper is well written and relatively easy to follow. The empirical evaluation does employ a set of strong baselines for comparison. The use of a "prototype bank" for backfilling is new to this reviewer and represents an interesting practical way to tackle missing values.

**Weaknesses:**

The paper introduces a few new key components, notably the prototype bank and the MDS-S4 architecture which, while well described, are only subjected to limited analysis and theoretical justification. For instance, complexity analysis is missing and ablation studies are partial. Some architectural choices seem arbitrary. The datasets chosen in the empirical evaluation (Traffic, Electricity, ETTh1, Weather) are all fairly simple datasets. Given that there are many more publicly available time-series evaluation datasets, I would like to see a more comprehensive evaluation.

**Questions:**

What is the computational complexity of ATPM vs traditional approaches?
Can you provide theoretical justification for the prototype bank design?
How sensitive is performance to prototype bank initialization?

---

> ### Author Response · Authors · 2024-11-25
> **Response Part 1**
>
> We thank the reviewer for the thoughtful and detailed review. Also, we appreciate that the reviewer acknowledges our prototype bank presents an interesting practical way to tackle missing values. We address the raised concerns below.
>
> ***1. What is the computational complexity of ATPM vs traditional approaches?***
>
> Thanks for the question, we provide the complexity analysis below. An **empirical comparison** of the computational costs of the S4M and other methods is given in the table below. S4M demonstrates **superior efficiency** in both training and inference compared to other baselines.
> | Method | S4(Mean) | S4(Ffill) | S4(Decay) | BRITS | GRUD | Transformer | Autoformer | BiTGraph | iTransformer | PatchTST | CRUD | Grafiti | S4M(Our) |
> | --- | --- | --- | --- | --- | --- | --- | --- | --- | --- | --- | --- | --- | --- |
> | Flops(M) | 12463.39 | 12463.39 | 12618.52 | 9091.16 | 3813.82 | 17627.87 | 18734.88 | 3185.64 | 565.36 | 392299.02 | 219.57 | 265118.32 | 139191.88 |
> | Training Time(s) | 0.11282 | 0.11498 | 0.08325 | 0.46920 | 0.19958 | 0.09035 | 0.09613 | 0.24546 | 0.06744 | 0.46017 | 49.40020 | OOM | 0.219381 |
> | Inference Time(s) | 0.07416 | 0.07983 | 0.06152 | 0.21126 | 0.08756 | 0.06088 | 0.07662 | 0.08122 | 0.04009 | 0.16896 | 4.76765 | OOM | 0.099314 |
>
> We also provide the **complexity analysis** for the core steps in bank writing and reading operations.
>
> - **Bank Writing:** Given $B$  training data points in a batch, each with $L$ segments, as we detailed in Algorithm 1 on the page, the core operations in writing the prototype bank involves four main steps: (1) randomly selecting  $n$ out of $B \cdot L$ representations, which has a complexity of $O(n)$; (2) computing similarity between $n$ selected representations and $s$ centroids of dimension $R$, leading to  $O(n \cdot s \cdot R)$; (3) selecting the maximum similarity for each representation, which requires $O(n \cdot s)$; and (4) updating the clustering via a FIFO-based mechanism, with a cost of $O(n)$. Assuming standard operations for similarity computation and FIFO updates, the overall computational complexity is dominated by $O(n \cdot s \cdot R)$, reflecting the influence of the embedding dimension $R$ and the number of centroids $s$
> - **Bank Reading:** Given $B$ training data points, each with $l$ segments, the procedure involves the following steps: (1) for all $B \cdot l$ segments, compute cosine similarity with $s$ centroids, resulting in a complexity of $O(B \cdot l \cdot s \cdot R)$, where $R$ is the embedding dimension; (2) select the top $K$ centroids for each segment, which takes $O(B \cdot l \cdot s)$ using a partial sort; (3) normalize the similarity values for these $K $ centroids using an exponential function, costing $O(B \cdot l \cdot K)$; and (4) compute the weighted average of these $K$ centroids, which also takes $O(B \cdot l \cdot K \cdot R)$. The overall computational complexity is dominated by $O(B \cdot l \cdot s \cdot R)$, primarily driven by the initial cosine similarity calculations.

---

> ### Author Response · Authors · 2024-11-25
> **Response Part 2**
>
> ***2. Theoretical justification.***
>
> Thank you for the questions. The long-term dependency in S4 is achieved using the HiPPO matrix $A$, as shown in (1). This long-term dependency is evident because the current state can be expressed as a convolution of previous states, with the convolution kernels being polynomial in the HiPPO matrix $A$. Our dual-stream processing maintains the HiPPO structure, as described in line 273 of our manuscript. Specifically, the current state remains a convolution of previous states, with the convolution kernel being polynomial in the HiPPO matrix, just as in S4. Therefore, the theoretical results for S4 remain valid in our approach.
>
> ***3. Additional ablation studies.***
>
> Thank you for your suggestion on additional ablation studies. We have included an ablation study for the first module, ATPM. In this study, we compare S4M with and without ATPM, highlighting the improvements brought by ATPM, particularly on the Traffic and ETTh1 datasets. The following experiments were conducted under the same settings as those in the ablation studies presented in the paper.
>
> | Dataset |  | Electricity |  | ETTh1 |  | Weather |  | Traffic |  |
> | --- | --- | --- | --- | --- | --- | --- | --- | --- | --- |
> | $\ell_L$ | Metrics | S4M (Ours) | S4M (w/o prototype) | S4M (Ours) | S4M (w/o prototype) | S4M (Ours) | S4M (w/o prototype) | S4M (Ours) | S4M (w/o prototype) |
> |  |  |  |  | Variable Missing |  |  |  |  |  |
> | 96 | MAE | 0.369 | +0.011 | 0.571 | +0.044 | 0.336 | +0.020 | 0.442 | +0.024 |
> |  | MSE | 0.282 | +0.010 | 0.624 | +0.091 | 0.267 | +0.206 | 0.786 | +0.125 |
> | 192 | MAE | 0.357 | +0.010 | 0.568 | +0.045 | 0.320 | +0.600 | 0.381 | +0.030 |
> |  | MSE | 0.261 | +0.009 | 0.598 | +0.090 | 0.261 | +0.002 | 0.685 | +0.092 |
> | 384 | MAE | 0.359 | +0.009 | 0.584 | +0.029 | 0.334 | +0.006 | 0.383 | +0.026 |
> |  | MSE | 0.264 | +0.009 | 0.613 | +0.064 | 0.256 | +0.008 | 0.700 | +0.065 |
> | 768 | MAE | 0.362 | +0.020 | 0.599 | +0.028 | 0.341 | +0.016 | 0.383 | +0.026 |
> |  | MSE | 0.269 | +0.002 | 0.649 | +0.058 | 0.266 | +0.011 | 0.697 | +0.074 |
> |  |  |  |  | Timepoint Missing |  |  |  |  |  |
> | 96 | MAE | 0.372 | +0.025 | 0.571 | +0.049 | 0.313 | +0.021 | 0.428 | +0.045 |
> |  | MSE | 0.287 | +0.030 | 0.624 | +0.110 | 0.237 | +0.017 | 0.809 | +0.116 |
> | 192 | MAE | 0.367 | +0.004 | 0.574 | +0.039 | 0.305 | +0.006 | 0.385 | +0.005 |
> |  | MSE | 0.274 | +0.004 | 0.593 | +0.110 | 0.225 | +0.001 | 0.687 | +0.023 |
> | 384 | MAE | 0.370 | +0.014 | 0.571 | +0.057 | 0.306 | +0.012 | 0.385 | +0.013 |
> |  | MSE | 0.277 | +0.004 | 0.624 | +0.112 | 0.220 | +0.015 | 0.702 | +0.047 |
> | 768 | MAE | 0.373 | +0.013 | 0.588 | +0.048 | 0.316 | +0.005 | 0.388 | +0.000 |
> |  | MSE | 0.282 | +0.016 | 0.647 | +0.079 | 0.232 | +0.004 | 0.699 | +0.024 |

---

> ### Author Response · Authors · 2024-11-25
> **Response Part 3**
>
> ***4. More comprehensive evaluation.***
>
> The datasets for empirical evaluation are widely used in time series forecasting literature [2,3]. We selected these four datasets because they exhibit significant variation in size, number of variables, and the presence or absence of seasonality. We consider cases with block missing patterns in regularly sampled time series, where the observed values occur at consecutive time points ((see our Fig. 4). This structure enables the design of an informative representation $o_t$, which is crucial for capturing temporal dependencies effectively.
>
> Following your comment, we included the real-world USHCN climate dataset [1] in our analysis. We set the lookback window of size 96 and a horizon window of size 96. The results further confirm that **S4M outperforms** other methods on the real-world dataset.
>
>
> | r | Metric | S4(Mean) | S4(Ffill) | S4(Decay) | BRITS | GRUD | Transformer | Autoformer | BiTGraph | iTransformer | PatchTST | CARD | S4M(Ours) |
> | --- | --- | --- | --- | --- | --- | --- | --- | --- | --- | --- | --- | --- | --- |
> | 0.12 | MAE | 0.477 | 0.489 | 0.466 | 0.644 | 0.477 | 0.461 | 0.511 | 0.474 | 0.478 | 0.494 | 0.451 | 0.447 |
> |  | MSE | 0.455 | 0.414 | 0.447 | 0.668 | 0.452 | 0.406 | 0.499 | 0.439 | 0.460 | 0.457 | 0.411 | 0.417 |
> | 0.24 | MAE | 0.507 | 0.522 | 0.502 | 0.644 | 0.499 | 0.475 | 0.534 | 0.495 | 0.504 | 0.528 | 0.477 | 0.473 |
> |  | MSE | 0.503 | 0.517 | 0.503 | 0.689 | 0.484 | 0.403 | 0.530 | 0.469 | 0.502 | 0.502 | 0.444 | 0.433 |
>
> Reference:
>
> [1] Long-term daily climate records from stations across the contiguous united states, 2015.
>
> [2] Autoformer: Decomposition transformers with auto-correlation for long-term series forecasting. In Advances in Neural Information Processing Systems, 2021
>
> [3] CARD: Channel aligned robust blend transformer for time series forecasting[C]. The Twelfth International Conference on Learning Representations. 2024.
>
> ***5. How sensitive is performance to prototype bank initialization?***
>
> In our experiment, we found the performance is **insensitive** to the initial cluster configuration, as the clusters are updated continuously throughout the training process. To provide evidence, we presented the experimental results on four datasets using different cluster numbers for initialization, as shown below. In practice, we recommend using 3 to 5 clusters for initialization or determining the optimal number of clusters based on the within-cluster sum of squares.
> |  | num_cluster | 1 | 2 | 3 | 4 | 8 | 12 | 16 |
> | --- | --- | --- | --- | --- | --- | --- | --- | --- |
> | Electricity | MAE | 0.415 | 0.415 | 0.415 | 0.415 | 0.415 | 0.415 | 0.415 |
> |  | MSE | 0.356 | 0.356 | 0.356 | 0.356 | 0.357 | 0.358 | 0.356 |
> | ETTh1 | MAE | 0.647 | 0.647 | 0.648 | 0.648 | 0.647 | 0.648 | 0.650 |
> |  | MSE | 0.768 | 0.767 | 0.770 | 0.770 | 0.767 | 0.767 | 0.773 |
> | Weather | MAE | 0.386 | 0.390 | 0.387 | 0.385 | 0.388 | 0.388 | 0.385 |
> |  | MSE | 0.307 | 0.310 | 0.308 | 0.306 | 0.310 | 0.310 | 0.307 |
> | Traffic | MAE | 0.510 | 0.505 | 0.504 | 0.515 | 0.509 | 0.513 | 0.509 |
> |  | MSE | 0.966 | 0.954 | 0.944 | 0.999 | 0.974 | 0.992 | 0.985 |

---

> ### Author Response · Authors · 2024-11-28
>
> Dear reviewer iDLq,
>
> We sincerely thank you for your thoughtful review and valuable feedback. We have carefully addressed each of your questions and provided detailed responses in the rebuttal. We hope to have resolved all your concerns. If you have any further comments, we would be glad to address them before the rebuttal period ends. If our responses address your concerns, we would deeply appreciate it if you could consider raising your score. Your recognition for our novel work means a lot. Thanks again for your time and effort in reviewing our work.
>
> Regards,
> S4M authors

---

> ### Comment · Area_Chair_Dm2o · 2024-12-02
>
> Dear Reviewer iDLq,
>
> You have indicated that submission 10665 is marginally below acceptance. The authors have provided a detailed response.
>
> Please indicate the extend to which their response addresses your concerns and explain your decision to update (or not update) your score.
>
> All the best,
>
> The AC

---

### Official Review · Reviewer_UFK6 · 2024-11-13

**Soundness:** 2
**Presentation:** 3
**Contribution:** 2
**Rating:** 5
**Confidence:** 3

**Summary:**

The paper presents S4M, an innovative end-to-end framework consisting of the Adaptive Temporal Prototype Mapper (ATPM) and the Missing-Aware Dual Stream S4 (MDS-S4) for multivariate time series forecasting that addresses the challenge of missing data.

**Strengths:**

1. The proposed S4M model is innovative, integrating missing data handling within the model architecture.

**Weaknesses:**

1. Although the results are promising, the authors did not provide their source code which results in very low reproducibility;
2. Computational efficiency comparison is missing;

**Questions:**

1. What are the computational costs and scalability of the S4M model, especially when dealing with large-scale multivariate time series data with high missing ratios? How does it compare to the baseline models in terms of training and inference time?
2. How does the dual stream processing impact the model's ability to capture temporal dependencies?
3. Can the authors confirm whether they implemented these baseline methods using official code or leveraged existing unified Python libraries, such as the Time-Series-Library [1] or PyPOTS [2]? It's important to note that data processing varies significantly among different imputation algorithms. Utilizing unified interfaces could help ensure that the experimental comparisons are conducted fairly.

### References
[1] https://github.com/thuml/Time-Series-Library

[2] Wenjie Du. PyPOTS: a Python toolbox for data mining on Partially-Observed Time Series. In KDD MiLeTS Workshop, 2023. https://github.com/WenjieDu/PyPOTS

---

> ### Author Response · Authors · 2024-11-25
> **Response Part 1**
>
> We thank the reviewer for the thoughtful and detailed review. Also, we appreciate that the reviewer acknowledges our proposed S4M model is innovative. We address your concerns below.
>
> ***W1. Source code.***
>
> Thanks for the question, the code can be found at the [anonymous link](https://anonymous.4open.science/r/S4M-C3FA/README.md).
>
> ***W2 & Q1. Computational costs of S4M model and its comparison to the baseline models in terms of training and inference time?***
>
> Thank you for your valuable feedback. S4M demonstrates **superior efficiency** in both training and inference compared to other baselines. To evaluate the computational cost of S4M, we conducted experiments using the Electricity dataset under the highest missing ratio setting. The experiments were performed with a batch size of 16 and a hidden size of 512.
>
> We observe that S4M (ours) achieves a **lower FLOPS value** compared to other SOTA transformer-based methods, including Grafiti. Also,  S4M (ours) is similar to the S4-based methods. The results confirm our motivation to focus on S4-based architecture, given their efficiency (see lines 51-52 of our original submission). Furthermore, S4M demonstrates **shorter training** times than CRUD, PatchTST, BiTGraph, and BRITS. For inference, S4M also outperforms CRUD, PatchTST, and BRITS, making it a **more efficient** choice for both training and inference. (For clarification, "OOM" in the tables refers to "Out-of-Memory.")
> | Method | S4(Mean) | S4(Ffill) | S4(Decay) | BRITS | GRUD | Transformer | Autoformer | BiTGraph | iTransformer | PatchTST | CRUD | Grafiti | S4M(Our) |
> | --- | --- | --- | --- | --- | --- | --- | --- | --- | --- | --- | --- | --- | --- |
> | Flops(M) | 12463.39 | 12463.39 | 12618.52 | 9091.16 | 3813.82 | 17627.87 | 18734.88 | 3185.64 | 565.36 | 392299.02 | 219.57 | 265118.32 | 139191.88 |
> | Training Time(s) | 0.11282 | 0.11498 | 0.08325 | 0.46920 | 0.19958 | 0.09035 | 0.09613 | 0.24546 | 0.06744 | 0.46017 | 49.40020 | OOM | 0.219381 |
> | Inference Time(s) | 0.07416 | 0.07983 | 0.06152 | 0.21126 | 0.08756 | 0.06088 | 0.07662 | 0.08122 | 0.04009 | 0.16896 | 4.76765 | OOM | 0.099314 |

---

> ### Author Response · Authors · 2024-11-25
> **Response Part 2**
>
> ***Q2. How does the dual stream processing impact the model's ability to capture temporal dependencies?***
>
> Thank you for the question. The motivation behind the dual-stream processing is to take advantage of the strengths of S4—namely, its ability to capture long-term temporal dependencies and its computational efficiency—when addressing block missing patterns in time series. The long-term dependency in S4 is achieved through the use of the HiPPO matrix $A$ as shown in (1). In our dual-stream processing, we build on this structure by incorporating $o_t$ (the representation from a shorter look-back window) instead of $u_t$ (the observation of only at the current time point) into the model. The term $o_t$ inherently captures additional temporal dependency information.
>
> ***Q3. Can the authors confirm whether they implemented these baseline methods using official code or leveraged existing unified Python libraries.***
>
> Thanks for checking the experiment details. For a fair comparison, we used the official implementation for all baselines.

---

> ### Author Response · Authors · 2024-11-28
>
> Dear reviewer UFK6,
>
> We sincerely thank you for your thoughtful review and valuable feedback. We have carefully addressed each of your questions and provided detailed responses in the rebuttal. We hope to have resolved all your concerns. If you have any further comments, we would be glad to address them before the rebuttal period ends. If our responses address your concerns, we would deeply appreciate it if you could consider raising your score. Your recognition for our novel work means a lot. Thanks again for your time and effort in reviewing our work.
>
> Regards,
> S4M authors

---

> ### Comment · Area_Chair_Dm2o · 2024-12-02
> **Please respond to the authors of submission 10665**
>
> Dear Reviewer UFK6,
>
> The discussion period is almost over, so please read the response of the authors of submission 10665 to your review.
>
> Does their response address your concerns? Will you modify your score? Please explain your decision.
>
> All the best,
>
> The AC

---

### Author Response · Authors · 2024-11-25
**Sincere thanks to all the reviewers**

We sincerely thank all reviewers for their time and valuable feedback. We are thrilled that the reviewers recognized the strengths of our work, describing our approach as “practical” and “innovative” and noting that the “idea of the prototype bank is very compelling and thoughtful.” We are also encouraged by the positive comments on the experimental results, which were described as “promising” and comprehensive, with “a lot of ablation studies/hyper-parameter experiments.” Additionally, we appreciate the acknowledgment that our paper is “well written” and “easy to follow.” We have carefully addressed your comments point by point. We appreciate the time and effort you have put into your review, and we welcome any further questions you may have.

---

### Meta-Review · Area_Chair_Dm2o · 2024-12-21

**Metareview:**

Most of the reviewers have appreciated the novelty of the method and its practical relevance, as well as the baselines chosen. Some design choices (such as the prototype bank) were deemed to be new solutions to the problem and of potential interest to the community.

There were concerns about the reproducibility of the method and its computational efficiency, raised by reviewer UFK6. The authors have shared their code and conducted experiments, showing their method does not introduce a large computational overhead compared to S4, and is competitive against baselines from the transformer family. The reviewer (who gave a score of borderline reject) did not participate in the discussion, even when prompted. However, I consider the issues they raised as having been addressed by the authors. There were no other reasons stated in the review as arguments to reject the paper.

An issue raised by Reviewer iDLq was the simplicity of the datasets and the need for more ablation studies. The authors have included a more complex dataset and additional ablation studies, which seemed to have convinced the reviewer since he raised his score. I also find these experiments a good addition to the paper.

Reviewer iyXX also appreciated the new ideas put forward in the paper as well as the authors’ response with additional experiments comparing against PatchTST and other models, which the reviewer found convincing.

Reviewer EZj4 raised some questions about modeling choices and hyperparameters, as well as the addition of experiments with various horizons. The authors have, in my option, addressed these issues, though the reviewer did not respond, even when prompted to do so.

All in all, there is sufficient novelty in this method to make it a valuable contribution to ICLR. Reviewers have requested additional experiments, which the authors provided, and which will also strengthen the paper. Thus, I recommend acceptance.

**Additional Comments On Reviewer Discussion:**

The meta-review contains a summary of the issues raised, and the author responses. The two reviewers who opted to marginally reject the paper did not participate in the discussion. I used my own judgement and determined that the authors addressed their comments.

---

### Decision · Program_Chairs · 2025-01-22

Accept (Poster)